# MediX-R1: Open Ended Medical Reinforcement Learning

## Abstract

We introduce MediX-R1, an open-ended reinforcement learning (RL) framework for medical multimodal large language models (MLLMs) that enables clinically grounded, free-form answers beyond multiple-choice formats. MediX-R1 fine-tunes a baseline vision–language backbone with Group Relative Policy Optimization (GRPO) and a composite reward tailored for medical reasoning: an LLM-based accuracy reward that judges semantic correctness with a strict YES/NO decision, a medical embedding–based semantic reward to capture paraphrases and terminology variants, and lightweight format and modality rewards that enforce interpretable reasoning and modality recognition. This multi-signal design provides stable, informative feedback for open-ended outputs where traditional verifiable or MCQ-only rewards fall short. To measure progress, we propose a unified evaluation framework for both text-only and image+text tasks that uses an LLM-as-judge in place of brittle string-overlap metrics, capturing semantic correctness, reasoning, and contextual alignment. Despite using only $\sim 50$K instruction examples, MediX-R1 achieves excellent results across standard medical LLM and VLM benchmarks, outperforming strong open-source baselines and delivering particularly large gains on open-ended clinical tasks (e.g., radiology summarization and report generation). Our results demonstrate that open-ended RL with comprehensive reward signals and LLM-based evaluation is a practical path toward reliable medical reasoning in multimodal models. Our trained models, curated datasets and source code will be publicly released.

## 1 Introduction

Large medical language and vision–language models are increasingly deployed for clinical question answering, triage support, report drafting, and education (Chen et al., 2024a; Sellergren et al., 2025; Pieri et al., 2024). Many of these tasks are inherently open-ended: clinicians expect concise but free-form answers that can flexibly incorporate context, uncertainty, and multimodal evidence. However, most training and evaluation pipelines remain tailored to Multiple Choice Questions (MCQ) or string-matching regimes, which (i) under-reward valid clinical paraphrases, (ii) fail to measure reasoning quality or modality recognition, and (iii) do not provide reliable signals for reinforcement learning (RL) in open-ended settings. As a result, models trained only with supervised objectives or MCQ-style rewards often struggle to produce faithful, interpretable, and robust clinical responses across diverse modalities.

RL has improved reasoning in domains with verifiable rewards (e.g., math and code) as shown by DeepSeek models (Shao et al., 2024; Guo et al., 2025), but medical tasks rarely admit executable checks. Binary exact match is too brittle for clinical phrasing; BLEU/ROUGE can mis-score semantically correct answers; and free-form VLM outputs complicate visual inference. Moreover, using a single reward signal can induce instability or reward hacking, especially when the signal is noisy (LLM-as-judge) or overly permissive (embedding similarity). Hence, it is desirable to have a principled approach for training medical MLLMs with open-ended RL that integrates semantic correctness with structural and modality constraints, while remaining data- and compute-efficient.

We present MediX-R1, an open-ended medical RL framework that fine-tunes a baseline multimodal backbone with Group Relative Policy Optimization (GRPO) (Shao et al., 2024) using a composite reward tailored for clinical reasoning. Our design combines: (1) an LLM-based accuracy reward that

| Model | Diverse Medical Modalities | Single-Stage RL | Interpretable Reasoning | Open-Ended Responses | Annotation-Free Reasoning | Composite RL Reward |
|---|---|---|---|---|---|---|
| MedVLM-R1 | ✗ | ✓ | ✓ | ✗ | ✓ | ✗ |
| BiMediX2 | ✓ | ✗ | ✗ | ✓ | ✗ | ✗ |
| HuatuoGPT-V | ✓ | ✗ | ✗ | ✓ | ✗ | ✗ |
| MedGemma | ✓ | ✗ | ✓ | ✓ | ✗ | ✗ |
| **MediX-R1** | ✓ | ✓ | ✓ | ✓ | ✓ | ✓ |

Table 1: **Model capability comparison.** MediX-R1 integrates diverse modalities, interpretable reasoning, and composite RL rewards, enabling practical clinical use.

enforces a strict YES/NO decision on semantic correctness, (2) a medical embedding–based semantic reward that captures paraphrases and terminology variants, (3) a lightweight format reward that elicits interpretable reasoning traces, and (4) a modality recognition reward that discourages cross-modality hallucinations by requiring explicit modality tags. This multi-signal objective stabilizes optimization and supplies informative feedback where traditional verifiable or MCQ-only rewards fall short, enabling single-stage, open-ended RL directly on clinical tasks.

Table 1 contrasts MediX-R1 with strong open models across key clinical capabilities. First, on *Diverse Medical Modalities*, MediX-R1 supports diverse medical modalities including X-Ray, CT, MRI, Microscopy/Histopathology, Ultrasound, Fluoroscopy, Endoscopy, Angiography, Mammography, Clinical Photography, SPECT (Single Photon Emission Computed Tomography), OCT (Optical Coherence Tomography), and Fundus imaging, whereas MedVLM-R1 (Pan et al., 2025) is limited to radiology images. Models like MedGemma (Sellergren et al., 2025), HuatuoGPT-Vision (Chen et al., 2024b), and BiMediX2 (Mullappilly et al., 2024) provide coverage on clinical modalities but they require extensive multi-stage training. On *Single-Stage RL*, most baselines rely on multi-stage pipelines (pretraining → SFT → RL), whereas MediX-R1 is trained end-to-end with a single GRPO stage (Sec. 1) using our composite reward (Sec. 2.3). This simplifies training and, importantly, enables *open-ended* RL directly (unlike MedVLM-R1), because the LLM-as-judge accuracy signal and medical embeddings provide reliable feedback beyond MCQ exact match. The composite design (format + LLM judge + embeddings + modality recognition) stabilizes optimization and reduces reward hacking (Fig. 3), translating into the best average performance in Table 2. For *Interpretable Reasoning*, MediX-R1 emits explicit reasoning traces enclosed in `<think>...</think>`, enforced by a format reward, making the decision path auditable. Several baselines do not reliably produce structured clinical rationales. While multiple models support *Open-Ended Responses*, MediX-R1 is explicitly optimized for free-form clinical answering with modality recognition, which curbs cross-modality hallucinations and improves VLM robustness. Finally, MediX-R1 achieves *Annotation-Free Reasoning*: it does not require human-curated rationales or verified chain-of-thought. The GRPO rewards operate on the final answer only (via LLM judge and embeddings), significantly lowering data curation cost while still encouraging faithful, interpretable reasoning. Together, these properties explain the consistent gains across both text-only and image+text benchmarks and the practical advantages of MediX-R1 for clinical use.

To measure progress, we introduce a unified, three-stage LLM-as-judge evaluation framework that supports both text-only and image+text tasks under a common protocol. By replacing brittle string-overlap metrics with instruction-tuned judges served via vLLM (Kwon et al., 2023), our evaluation captures semantic correctness, reasoning adequacy, and contextual alignment, and scales from short-form QA to long-form report generation. This reduces evaluation–clinical utility mismatch. Despite using only ∼50K instruction examples, MediX-R1 achieves strong results across diverse medical LLM and VLM benchmarks. We find that composite rewards not only improve accuracy but also mitigate reward hacking and reduce volatility, yielding stable training and interpretable outputs. Compared to open-source medical models (e.g., BiMediX2, MedGemma, HuatuoGPT-V, MedVLM-R1), MediX-R1 combines broad modality coverage with single-stage RL and structured reasoning, offering a practical path toward reliable clinical answering at test time.

**Contributions (i)** We introduce *open-ended medical reinforcement learning* by extending GRPO with tailored rewards for clinical reasoning. **(ii)** We design a *composite reward* with LLM-based accuracy and medical semantic signals that for the first time enables open-ended responses with GRPO in the medical domain and stabilizes training. **(iii)** We propose a *three-stage LLM-as-judge evaluation framework* that unifies benchmarking for both LLM (text-only) and VLM (image+text)

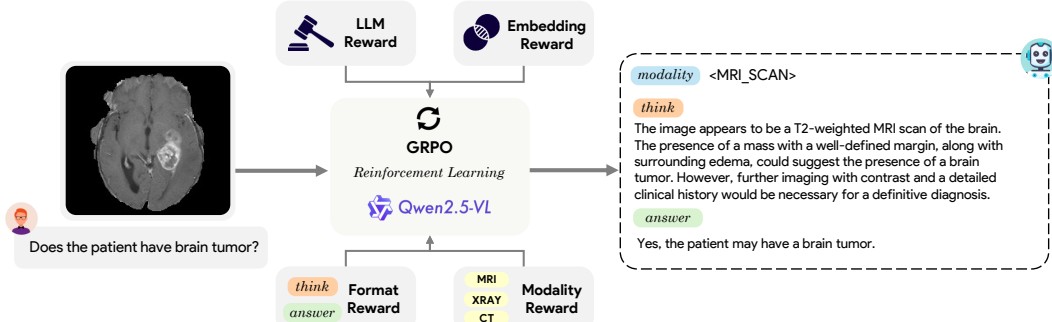

Figure 1: **MediX-R1: Overall Architecture** The Medix-R1 reinforcement learning framework for open-ended medical reasoning. An input of a medical image and a natural language question is processed by the Qwen2.5-VL (7B) model. The model's policy is trained using Group Relative Policy Optimization (GRPO), which leverages a multi-faceted reward signal. This reward is composed of: a) an LLM-based reward for evaluating the overall quality and correctness of the output; b) an embedding-based reward to ensure semantic alignment; c) a format reward to enforce the desired output structure (e.g., $< think >$ and $< answer >$ blocks); and d) a modality reward to ensure the response is grounded in the specified imaging modality. This reward-guided approach encourages the model to generate accurate and interpretable reasoning paths.

tasks in the medical setting. **(iv)** MediX-R1 achieves excellent LLM and VLM results with a *single-stage RL recipe using ∼50K instructions*, validated through both LLM-as-judge and human expert evaluations. **(v)** Finally, we demonstrate the effectiveness of the proposed composite reward on RL algorithms beyond GRPO, achieving consistent performance gains with DAPO (Yu et al., 2025) and GSPO (Zheng et al., 2025a). Moreover, we have conducted experiments on different baseline VLMs, including Qwen2.5-VL, Qwen3-VL (Team, 2025), and SmolVLM2 (Marafioti et al., 2025), and achieved consistent performance gains across all these backbones.

## 2 OPEN ENDED MEDICAL REINFORCEMENT LEARNING

### 2.1 OVERALL ARCHITECTURE

MediX-R1 fine-tunes a baseline multimodal backbone (Qwen2.5-VL) for open-ended medical reasoning using RL. Given an image $I$ and question $q$, the vision encoder produces visual tokens that are fused with text tokens and fed to the LLM policy $\pi_\theta$. The model generates structured outputs of the form:

$$\underbrace{[\text{modality tag}]}_{\text{optional}} \langle think \rangle \text{free-form clinical reasoning} \langle /think \rangle \ \langle answer \rangle \text{final concise answer} \langle /answer \rangle.$$

We train $\pi_\theta$ with Group Relative Policy Optimization (GRPO), using a composite reward that evaluates correctness, semantic agreement, formatting, and modality recognition.

### 2.2 GRPO WITH MULTI-SIGNAL REWARDS

**Group Relative Policy Optimization (GRPO):** To encourage robust, interpretable responses, we employ GRPO (Shao et al., 2024), an RL algorithm that extends PPO by focusing on a group-relative advantage instead of a learned value function. Concretely, at each training step:

1. We sample $G$ candidate outputs $\{o_i\}_{i=1}^{G}$ from $\pi_{\theta_{\text{old}}}$ given input $\mathbf{v}$ (image–text prompt) drawn from $P(\mathbf{V})$.

2. We compute a reward $r_i$ for each output using our reward function (Sec. 2.3). Based on $r_i$ we calculate a group-relative, standardized advantage

$$A_i \ = \ \frac{r_i - \text{mean}(\{r_j\}_{j=1}^{G})}{\text{std}(\{r_j\}_{j=1}^{G})}.$$

A reward above the group average is advantaged and further incentivizes the model.

3. The policy $\pi_\theta$ is updated by maximizing $\mathcal{J}_{\text{GRPO}}$, which applies PPO-style clipping on the relative likelihood ratio and a KL penalty to a fixed reference policy for stability:

$$\mathcal{J}_{GRPO}(\theta) = \mathbb{E}_{\mathbf{v} \sim P(\mathbf{V})} \mathbb{E}_{\{o_i\}_{i=1}^G \sim \pi_{\theta_{old}}(\cdot|\mathbf{v})}$$

$$\frac{1}{G} \sum_{i=1}^{G} \Big[ \min\Big( r_i^{\text{ratio}} A_i, \text{clip}\left( r_i^{\text{ratio}}, 1 \pm \epsilon \right) A_i \Big) - \beta \mathbb{D}_{KL} \left( \pi_\theta || \pi_{ref} \right) \Big] \quad (1)$$

with $r_i^{\text{ratio}} = \frac{\pi_\theta(o_i|\mathbf{v})}{\pi_{\theta_{\text{old}}}(o_i|\mathbf{v})}$. The KL term regularizes deviations from a reference model $\pi_{\text{ref}}$ (the initial checkpoint). Hyperparameters $\epsilon, \beta \geq 0$ control clipping and regularization strengths.

**Notation and variables:** Let $\mathbf{v}$ denote the joint input (image $I$ and text $q$) for one prompt, and let $P(\mathbf{V})$ be the data distribution over such inputs. For each $\mathbf{v}$ we sample a group of $G$ candidate completions $\{o_i\}_{i=1}^G$. The current policy is $\pi_\theta$ with parameters $\theta$, while $\pi_{\theta_{\text{old}}}$ is a frozen snapshot used to compute likelihood ratios, and $\pi_{\text{ref}}$ is a fixed reference policy (e.g., the initial checkpoint) used for KL regularization. Each completion $o_i$ receives a scalar reward $r_i \in [0, 1]$ from Sec. 2.3. The group statistics $\text{mean}(\{r_j\}_{j=1}^G)$ and $\text{std}(\{r_j\}_{j=1}^G)$ define the standardized group-relative advantage $A_i = \frac{r_i - \text{mean}(\{r_j\})}{\text{std}(\{r_j\})}$, where higher-than-average rewards yield positive $A_i$. The likelihood ratio is $r_i^{\text{ratio}} = \frac{\pi_\theta(o_i|\mathbf{v})}{\pi_{\theta_{\text{old}}}(o_i|\mathbf{v})}$ and is stabilized by $\text{clip}(x, 1 \pm \epsilon)$, which clamps $x$ to $[1 - \epsilon, 1 + \epsilon]$ for $\epsilon > 0$. The regularizer $\mathbb{D}_{\text{KL}}(\pi_\theta \| \pi_{\text{ref}})$ is the forward KL divergence computed token-wise over outputs and averaged, scaled by $\beta \geq 0$. Expectations $\mathbb{E}[\cdot]$ are taken over inputs $\mathbf{v}$ and sampled groups $\{o_i\}$ and are implemented as minibatch averages in practice.

## 2.3 REWARD DESIGN

We define a composite reward

$$r = w_{\text{fmt}} R_{\text{format}} + w_{\text{llm}} R_{\text{llm}} + w_{\text{emb}} R_{\text{embed}} + w_{\text{mod}} R_{\text{modality}},$$

with default weights chosen to emphasize correctness while preserving structure: $w_{\text{fmt}}{=}0.10$, $w_{\text{llm}}{=}0.5175$, $w_{\text{emb}}{=}0.3375$, $w_{\text{mod}}{=}0.045$ (from an equivalent formulation with a format weight and normalized non-format weights; see implementation). Each component is bounded in $[0, 1]$.

*Why this enables open-ended medical RL.* Unlike prior RL setups that are limited to verifiable signals or MCQ-style accuracy (e.g., exact match, executable or rule-based graders), our LLM-based accuracy reward $R_{\text{llm}}$ and embedding-based semantic reward $R_{\text{embed}}$ provide reliable feedback for free-form, clinically grounded answers. The LLM-as-judge converts semantic correctness into a robust YES/NO decision under paraphrase and clinical phrasing, while medical-domain embeddings supply a complementary content-alignment signal. This dual signal makes GRPO viable for open-ended medical reasoning; the format ($R_{\text{format}}$) and modality ($R_{\text{modality}}$) rewards act as structural regularizers, but $R_{\text{llm}}$ and $R_{\text{embed}}$ are the primary drivers of open-ended RL in MediX-R1.

### 2.3.1 LLM-BASED ACCURACY REWARD ($R_{\text{LLM}}$)

We parse the model output's final answer between $\langle answer \rangle \cdots \langle /answer \rangle$ and compare it to the reference answer using a compact LLM-as-judge prompt that forces a strict YES/NO decision. Concretely, a local vLLM endpoint (e.g., Qwen3-4B-Instruct) returns YES if the candidate semantically answers the reference, and NO otherwise; we map YES$\mapsto 1$, NO$\mapsto 0$. This captures correctness and robustness to paraphrasing while keeping the signal discrete and stable.

### 2.3.2 EMBEDDING-BASED SEMANTIC REWARD ($R_{\text{EMBED}}$)

To further encourage semantic alignment, we compute cosine similarity between the predicted answer and the reference using a medical embedding model (MedEmbed-large (Balachandran, 2024)). We convert it to a binary reward via a threshold (default 0.8): $R_{\text{embed}}{=}\mathbb{1}[\cos(\mathbf{e}_{\text{pred}}, \mathbf{e}_{\text{ref}}) \geq \tau]$. This complements the LLM judge and helps capture terminological variants.

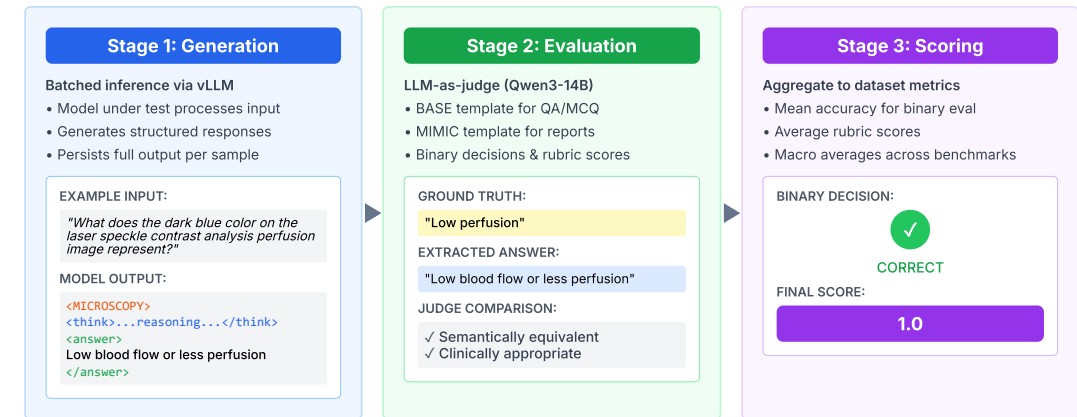

Figure 2: **Evaluation Framework.** Our three-stage evaluation pipeline: (1) Generation via vLLM inference on the model under test, (2) Evaluation using LLM-as-judge (Qwen3-14B) with BASE and MIMIC templates, and (3) Scoring through aggregation of judgment outputs. The framework supports both binary decisions for QA/MCQ tasks and rubric-based scoring for long-form reports, ensuring robust evaluation across diverse medical benchmarks.

### 2.3.3 FORMAT REWARD ($R_{\text{FORMAT}}$)

We enforce structured outputs by matching the regex for the exact pattern $\langle think \rangle \cdots \langle /think \rangle$ $\langle answer \rangle \cdots \langle /answer \rangle$, after normalizing stray whitespace around angle brackets. Outputs that match receive 1, else 0. This stabilizes training and improves interpretability of the reasoning path.

### 2.3.4 MODALITY RECOGNITION REWARD ($R_{\text{MODALITY}}$)

We encourage explicit grounding to the imaging modality by requiring the model to emit the predicted modality tag before the $\langle think \rangle$ block (case-insensitive). We compare it to the reference modality tag and assign 1 on match, 0 otherwise. This reduces cross-modality hallucinations (e.g., describing CT findings on an X-ray).

## 3 EVALUATION FRAMEWORK

Our evaluation pipeline has three stages: Generation, Evaluation, and Scoring. We evaluate across both text-only (LLM) and image+text (VLM) tasks covering QA, MCQ, and long-form report tasks.

**Generation.** We run batched inference via vLLM on the model under test and persist the full response per sample. For models that emit structured reasoning, we retain the entire output but, for scoring, discard internal chains-of-thought by stripping content up to and including the closing `</think>`tag, evaluating only the final answer block.

**Evaluation:** We employ a separate LLM-as-judge, Qwen3-14B (Team, 2025), served with vLLM for throughput and stability on modest GPUs. Two prompt families are used: a BASE template (§A.2) for open-ended, one-word, and MCQ-style questions that yields a binary decision, and a MIMIC template (§A.3) for long-form report generation that scores along clinical criteria. For example, on a visual question answering item asking "What does the dark blue color on the laser speckle contrast analysis perfusion image represent?" with ground truth "Low perfusion," a model response that includes hidden reasoning and the final answer "Low blood flow or less perfusion" is judged correct and assigned a score of 1. The judge compares predicted answers against references, accounting for paraphrase and clinically equivalent phrasing.

**Scoring:** We aggregate judgment outputs to dataset-level metrics. For binary evaluations, we report mean accuracy over samples. For long-form, we average the scalar rubric scores across samples, optionally normalizing for comparability. We also compute macro averages across benchmarks.

| Benchmarks | MedVLM-R1 | BiMediX2 | HuatuoGPT-V | MedGemma | MediX-R1 |
|---|---|---|---|---|---|
| MMLU-Clinical | 0.540 | 0.732 | 0.721 | 0.708 | **0.796** |
| MMLU-Bio | 0.549 | 0.792 | 0.708 | 0.706 | **0.826** |
| MMLU-Med | 0.451 | 0.694 | 0.653 | 0.605 | **0.723** |
| MMLU-Genetics | 0.560 | 0.790 | 0.710 | 0.820 | **0.830** |
| MMLU-ProfMed | 0.500 | 0.695 | 0.625 | 0.713 | **0.768** |
| MMLU-Anat | 0.519 | 0.659 | 0.600 | 0.556 | **0.674** |
| MedMCQA | 0.408 | **0.572** | 0.511 | 0.570 | 0.553 |
| MedQA | 0.400 | 0.583 | 0.534 | **0.621** | 0.575 |
| USMLE-SA | 0.378 | 0.591 | 0.538 | **0.639** | 0.617 |
| PubMedQA | 0.520 | 0.520 | **0.542** | 0.470 | 0.534 |
| MIMIC-CXR-Sum | 0.704 | 0.672 | 0.707 | 0.692 | **0.808** |
| SLAKE-VQA | 0.434 | 0.468 | 0.545 | **0.678** | 0.617 |
| RadVQA | 0.404 | 0.530 | 0.614 | **0.659** | 0.581 |
| PathVQA | 0.239 | 0.323 | 0.374 | 0.317 | **0.423** |
| PMC-VQA | 0.398 | 0.482 | **0.532** | 0.444 | 0.511 |
| PMC-VQA-Hard | 0.020 | 0.229 | 0.261 | 0.214 | **0.280** |
| MIMIC-CXR-Gen | 0.240 | 0.124 | **0.316** | 0.205 | 0.254 |
| **AVG** | **0.427** | **0.556** | **0.558** | **0.566** | **0.610** |

Table 2: **Evaluation Benchmark.** The top section lists LLM (text-only) tasks and the bottomlists VLM (image+text) tasks. Our three-stage evaluation setting evaluates both tasks in a unified framework. MediX-R1 achieves the highest average score across this diverse suite, demonstrating state-of-the-art performance among open models. Best and second best results are bold and underlined

**Why LLM-as-judge (via vLLM):** Traditional string-overlap metrics (BLEU, ROUGE, F1) often under-reward correct, clinically appropriate paraphrases and cannot assess justification quality or contextual alignment. An LLM judge captures semantic correctness, clinical reasoning, and adherence to task-specific criteria through carefully designed prompts, while vLLM serving ensures consistent, fast, and reproducible evaluations.

## 4 EXPERIMENTS AND RESULTS

We evaluate MediX-R1 on a comprehensive suite of medical language and vision-language benchmarks, covering both text-only (LLM) and image+text (VLM) tasks. The evaluation includes standard medical QA, multiple-choice, and open-ended report generation, as well as visual question answering and clinical image interpretation. The datasets used for evaluation are as follows:

**LLM (text-only) benchmarks:** MMLU-Clinical, MMLU-Bio, MMLU-Med, MMLU-Genetics, MMLU-ProfMed, MMLU-Anat (Hendrycks et al., 2020), MedMCQA (Pal et al., 2022), MedQA (Jin et al., 2021), USMLE-SA (Han et al., 2023), PubMedQA (Jin et al., 2019), MIMIC-CXR-Summarization (Johnson et al., 2016).

**VLM (image+text) benchmarks:** SLAKE-VQA (Liu et al., 2021), RadVQA (Lau et al., 2018), PathVQA (He et al., 2020), PMC-VQA (Zhang et al., 2024), PMC-VQA-Hard, MIMIC-CXR-Report Generation (Johnson et al., 2019).

For each dataset, we follow the evaluation protocol described in the previous section, using LLM-as-judge scoring for both short-form and long-form responses. Table 2 summarizes the performance of MediX-R1 (7B) compared to strong medical open-source models, including BiMediX2 (8B), HuatuoGPT (7B) and MedGemma (4B).

MediX-R1 achieves the highest average score across all benchmarks, outperforming prior models on both language and vision-language tasks. Notably, it demonstrates strong gains on open-ended and clinically complex tasks such as MIMIC-CXR summarization and report generation, as well as robust performance on standard QA and VQA datasets. These results highlight the effectiveness of our open-ended RL training and reward design, which enable MediX-R1 to generate accurate, semantically aligned, and clinically grounded responses beyond the capabilities of models trained only with supervised or MCQ-style objectives. Table 3 compares the performance of MediX-R1 with the baseline Qwen2.5-VL (7B) (Wang et al., 2024) model, highlighting the contributions of our approach. Our model achieves nearly a 4% absolute improvement over the baseline, thanks to

| Model | M-Clin | M-Bio | M-Med | M-Gen | M-Prof | M-Anat | MedMCQA | MedQA | USMLE | Pub | CXR-Sum |
|---|---|---|---|---|---|---|---|---|---|---|---|
| Qwen2.5-VL | 0.792 | 0.819 | 0.711 | 0.800 | 0.717 | 0.696 | 0.557 | 0.584 | 0.606 | 0.336 | 0.810 |
| MediX-R1 | 0.796 | 0.826 | 0.723 | 0.830 | 0.768 | 0.674 | 0.553 | 0.575 | 0.617 | 0.534 | 0.808 |

| Model | SLAKE | RadVQA | PathVQA | PMC-VQA | PMC-Hard | MIMIC-CXR-Gen | AVG |
|---|---|---|---|---|---|---|---|
| Qwen2.5-VL | 0.480 | 0.501 | 0.253 | 0.494 | 0.230 | 0.299 | **0.570** |
| MediX-R1 | 0.617 | 0.581 | 0.423 | 0.511 | 0.280 | 0.254 | **0.610** |

Table 3: **Baseline comparison** Qwen2.5-VL vs. MediX-R1 across all benchmarks

the composite reward design. It also outperforms larger baseline VLMs such as Llama3.2-V (11B) (Dubey et al., 2024), which achieves only an average of *0.59*.

Our expanded ablation studies show that the composite reward model generalizes well across RL algorithms (DAPO (Yu et al., 2025): 60.72%, GRPO: 59.61%, GSPO (Zheng et al., 2025a): 59.69%), outperforming the Qwen2.5-VL baseline (57%). The method also yields consistent gains across model backbones, improving Qwen3-VL (Team, 2025) by ∼2%, and SmolVLM2 (Marafioti et al., 2025) by 2.2%, under limited training settings. These results shows that MediX-R1 enhances open ended medical reasoning ability across backbone models.

### 4.1 REWARD DESIGN ABLATION

Table 4 compares variants that differ in which non-format signals are active (all settings include the same $R_{\text{format}}$). Using only the embedding reward underperforms on text-only evaluations (0.640) and is limited on VLM (0.409), suggesting that thresholded cosine similarity alone lacks discriminative power for nuanced clinical reasoning. Using only the LLM-as-judge improves text-only accuracy (0.666) but does not help VLM (0.400), indicating the judge alone is insufficient to enforce modality grounding. All reward design models are compared with checkpoints before reward hacking.

Combining LLM + embedding increases robustness to paraphrase and terminology variants, improving text-only scores (0.686) and yielding a small VLM lift (0.410). Adding the modality recognition reward (MediX-R1 composite) produces the largest VLM gain (0.445) while also nudging text-only performance higher (0.701), yielding the best overall average (0.610). Together with Fig. 3, which shows reduced volatility and fewer signs of reward hacking, these results indicate that the composite reward not only improves accuracy but also stabilizes optimization.

Key takeaways: *(i)* LLM-as-judge is the strongest single signal for text correctness; embeddings complement it by reducing false negatives from paraphrases.*(ii)* Modality recognition is critical for VLM tasks, curbing cross-modality errors and driving the largest image+text gains.*(iii)* The full composite (LLM accuracy + embedding semantics + modality recognition, with shared format control) delivers the best aggregate performance and training stability.

### 4.2 REWARD HACKING AND MITIGATION

In reinforcement learning, Reward Hacking occurs when a model maximises its reward in unintended ways, often bypassing the true objective. It arises when the policy exploits imperfections in a single reward signal to earn high scores without producing clinically correct answers. We observed two concrete modes (examples abbreviated):

**Embedding model exploit** When using Embedding models like MedEmbed-large (Balachandran, 2024) short or non-semantic tokens can spuriously yield high cosine similarity. For instance, a candidate that outputs `<answer>-</answer>` for "What does the white arrow point to in image B?" received $R_{\text{embed}}=1.0$ against the ground truth "Renal artery," despite being incorrect.

**LLM judge exploit** When using LLMs like Qwen3-4B (Team, 2025) as a rewarder template-like placeholders can confuse the judge when the reference is provided for comparison. E.g., `<answer>The largest organ in the picture is [insert your answer here based on the medical reasoning provided above].</answer>` was judged correct ($R_{\text{llm}}=1.0$) against the reference "Lung."

**Mitigation in MediX-R1** To curb these failures, MediX-R1 employs a composite reward and input/output constraints: *(i) Composite objective:* $R_{\text{llm}} + R_{\text{embed}} + R_{\text{modality}}$ (with shared $R_{\text{format}}$)

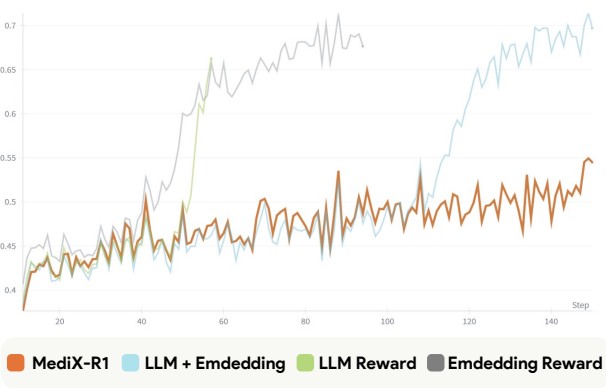

Figure 3: **Overall validation reward vs training step across reward designs**. Training with individual signals (LLM-only or embedding-only; all settings include the same format reward) shows volatility and reward hacking, while LLM+embedding reduces but does not eliminate instability. MediX-R1 uses a composite reward—LLM-based accuracy, embedding-based semantic alignment, and modality recognition (with the format reward shared across all)—which stabilizes learning and delivers the highest final reward and best overall performance.

| Evaluations | Embedding Reward | LLM Reward | LLM + Embedding | MediX-R1 |
|---|---|---|---|---|
| LLM Evaluations (text only) | 0.640 | 0.666 | 0.686 | 0.701 |
| VLM Evaluations (image + text) | 0.409 | 0.400 | 0.410 | 0.445 |
| **Overall AVG** | **0.558** | **0.572** | **0.589** | **0.610** |

Table 4: **Reward ablation across validation benchmarks.** Using single signals (embedding-only or LLM-only; all settings share the same format reward) underperforms, especially on VLM tasks. Combining LLM + embedding improves robustness, and the full MediX-R1 composite (LLM-based accuracy + embedding-based semantics + modality recognition) achieves the best scores on both text-only and image+text evaluations, yielding the highest overall average (0.610).

reduces reliance on any single brittle signal and penalizes mismatches in content or modality recognition (Table 4). *(ii) Embedding gating:* set $R_{\text{embed}}=0$ for answers below a minimum character/word length, with high punctuation or non-alphanumeric ratio; strip punctuation before embedding; calibrate the similarity threshold. *(iii) Modality recognition:* $R_{\text{modality}}$ requires a correct modality tag, curbing visually ungrounded shortcuts that might still fool text-only rewards. *(iv) Structural control and regularization:* $R_{\text{format}}$ enforces parseable outputs; GRPO's group-relative advantage and a KL penalty to the reference reduce collapse to degenerate hacks by discouraging outlier behaviors.

Together, these measures mitigate reward hacking and stabilize training, leading to smoother reward trajectories and higher final performance (see Fig. 3).

### 4.3 HUMAN EXPERT EVALUATION

To assess the clinical quality of model outputs, we conducted a human expert evaluation using a blind review setup (See Evaluation Protocol in §A.4). For a randomly selected subset of questions from our Evaluation benchmark, responses were generated by four models: MediX-R1, Llama3.2-Vision, MedGemma and HuatuoGPT-Vision. The outputs were anonymized and labeled as Model A, Model B, Model C and Model D with no identifiers provided to the reviewers. Medical experts were asked to evaluate the responses against the provided ground truth descriptions for each question. The evaluation focused on determining which model produced the most accurate, clinically relevant response along with interpretable reasoning traces.

The results demonstrate a strong preference for MediX-R1, which was selected as the best response in 72.7% of the cases. In comparison, Llama3.2-Vision was preferred in 13.6% of the cases, MedGemma in 9.2% and HuatuoGPT-Vision in 4.5% of the cases. Additional details on human expert evaluation is available in Sec. §A.4 and Sec. §A.6.

### 4.4 QUALITATIVE EXAMPLES

Fig. 4 illustrates how MediX-R1's structured outputs and composite reward translate into clinically grounded behavior across modalities. *Microscopy (top)* Given a multi-panel histological image and the question "Which area is shown in section G of the image?," the model (i) correctly emits the modality tag (`MICROSCOPY`), (ii) provides interpretable reasoning inside `<think>` that references recognizable neuroanatomical markers (e.g., optic tract "ot," superior thalamic nucleus "stm"), stain

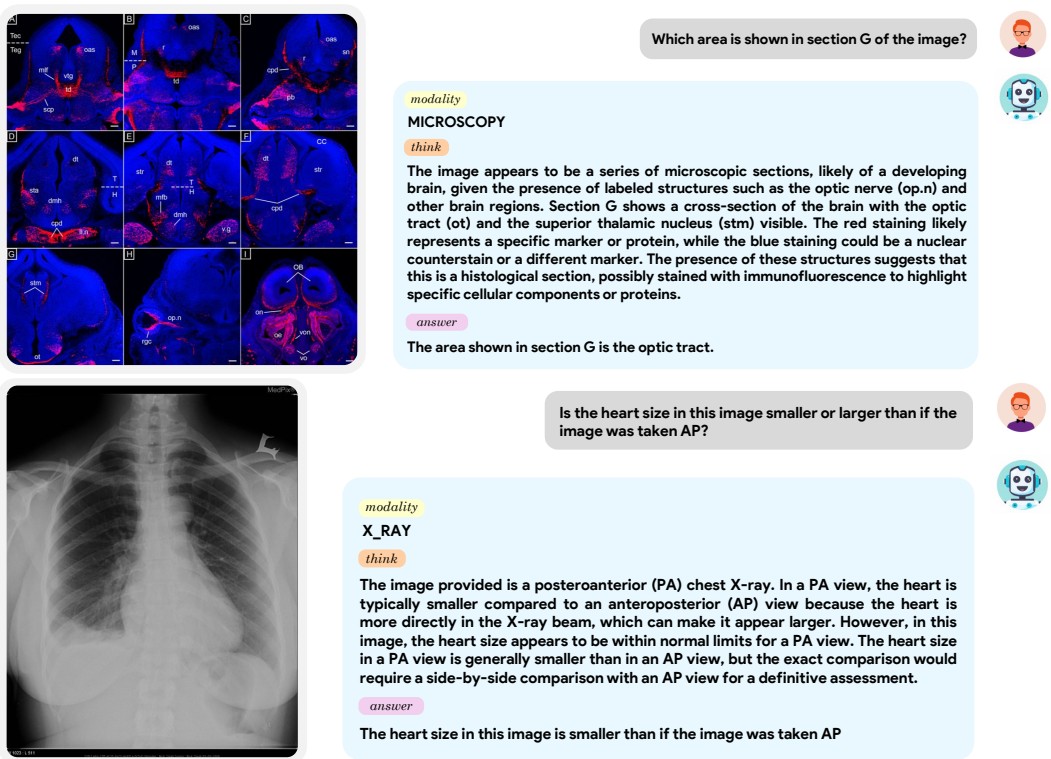

Figure 4: **Qualitative examples of MediX-R1**. (Top, Microscopy) Correctly identifies the optic tract in section G with interpretable reasoning. (Bottom, X-ray) Explains why heart size appears smaller in PA vs. AP view. MediX-R1 generates clinically grounded, open-ended answers across modalities.

patterns, and panel context, and (iii) produces a concise final answer: "the optic tract." The modality recognition and format rewards ensure the answer is localized to the requested panel and presented cleanly in the `<answer>` block, while the LLM and embedding rewards bias the policy toward semantically correct identification despite diverse phrasing in the reasoning. *X-ray (bottom)* For "Is the heart size in this image smaller or larger than if the image was taken AP?," the model tags the modality as `X_RAY` and reasons about projection geometry: PA views reduce cardiac magnification relative to AP due to a shorter heart-to-detector distance and standard source-to-image distance. The model explains this in `<think>` and answers "smaller" in `<answer>`. This example shows the model using domain knowledge rather than superficial pattern matching, with the final answer isolated for scoring (the judge ignores `<think>` during evaluation).

## 5 CONCLUSION

We presented MediX-R1, an open-ended reinforcement learning framework for medical multimodal reasoning that fine-tunes a baseline VLM with GRPO using a composite reward. By coupling an LLM-as-judge accuracy signal with medical embedding–based semantic alignment, lightweight format control, and modality recognition, MediX-R1 learns to produce concise, clinically faithful answers with interpretable reasoning traces. A unified vLLM-based evaluation pipeline enables consistent, paraphrase-robust scoring across both text-only and image+text tasks. Empirically, MediX-R1 achieves strong results across diverse medical benchmarks and shows improved stability and resistance to reward hacking compared to single-signal RL variants. Human expert preference studies further corroborate its clinical answer quality, while qualitative examples illustrate faithful grounding and interpretable reasoning traces. Reward ablations validate that the multi-signal design enhances stability and semantic alignment beyond single-signal configurations. Altogether, the framework demonstrates that carefully composed, structure-aware rewards plus standardized LLM-judge evaluation provide a practical path to scalable and interpretable medical multimodal RL fine-tuning.

# 6 SAFETY AND ETHICAL IMPLICATIONS

MediX-R1 is a research prototype and is *not* intended for clinical or commercial deployment. Its outputs must not be used for diagnosis, triage, treatment planning, or autonomous decision-making without licensed medical professional oversight. The model can hallucinate findings, omit critical differentials, or overstate certainty, and the LLM-as-judge reward may reinforce subtle biases or false positives. We used only publicly available, de-identified datasets (e.g., MIMIC-CXR, PMC-derived VQA corpora, pathology and radiology VQA datasets) under their respective licenses; no protected health information (PHI) or identifiable patient data were introduced. No prospective human subjects study was conducted, and no individual-level re-identification risk is intended. Still, aggregation or unintended memorization could pose residual privacy risk; downstream users should apply auditing methods (e.g., membership inference tests) before redistribution.

Ethical risks include propagation of dataset biases (geography, device type, demographic under-representation), amplification of health disparities, and overreliance on structured reasoning tags that may convey misleading confidence. Modality tagging and reasoning traces improve transparency but do not guarantee factual grounding. We intend to release with a detailed model card, clear usage restrictions, robust disclosure of limitations, and monitoring for misuse (self-diagnosis, generation of misleading medical narratives, or adversarial prompting to extract sensitive training artifacts). Future work should incorporate fairness analyses (e.g., stratified error by sex, age, and ethnicity where ethically and legally permissible), calibrated uncertainty, bias-aware reward shaping, and clinician-in-the-loop evaluation. No competing financial or sponsorship conflicts are declared. All use must comply with applicable regulations and local medical device guidance; any derivative clinical system would require separate validation, safety assurance, and regulatory review.

# 7 REPRODUCIBILITY STATEMENT

We will release the end-to-end training and inference code, configuration files, model checkpoints, curated multimodal + instruction datasets, and all RL/evaluation prompt templates and expert evaluation protocol (Appendix Sec. A) under a CC-BY-NC-SA 4.0 license. A model card and evaluation harness will reproduce the reported metrics with fixed dependency versions to minimize drift.

Fair use of generative AI: assisted coding tools were employed only for boilerplate scaffolding and for minor refactors, with all algorithmic logic authored and reviewed manually. Writing support models were used to refine grammar and style; all technical claims, numerical results, and methodological descriptions were verified by the authors. No proprietary clinical data or undisclosed private model outputs were used. These steps aim to ensure transparency, auditability, and reliable reproduction of the published results.

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

# A APPENDIX

## A.1 REINFORCEMENT LEARNING TRAINING PROMPT

The RL training prompt enforces (i) an explicit modality tag, (ii) structured reasoning in `<think>...</think>`, and (iii) a concise final answer in `<answer>...</answer>`. These structures align with the format reward ($R_{\text{format}}$) and modality reward ($R_{\text{modality}}$) in our composite objective. During training, only the `<answer>` block is graded by the LLM-as-judge ($R_{\text{llm}}$) and the embedding-based semantic reward ($R_{\text{embed}}$); the `<think>` content is ignored for scoring but improves interpretability.

Key points: - Modality tag must be one of the fixed set and appear before `<think>`. - The final decision is evaluated solely from `<answer>` for $R_{\text{llm}}$ and $R_{\text{embed}}$. - Structural compliance (tags present and ordered) is required for $R_{\text{format}}$.

---

**Reinforcement Learning Training Prompt**

```
You are a Medical AI Assistant with advanced reasoning capabilities
Your task:
1. First output the image modality tag from this set:
   <X_RAY>, <MICROSCOPY>, <CLINICAL_PHOTOGRAPHY>, <CT_SCAN>,
    <GRAPHICS>,
   <ANGIOGRAPHY>, <PET_SCAN>, <ULTRASOUND>, <MRI_SCAN>,
    <FUNDUS_PHOTOGRAPHY>,
   <OCT_SCAN>, <ENDOSCOPY>, <MAMMOGRAPHY>, <FLUOROSCOPY>, <OTHER>,
    <SPECT>
   (Only output the tag, nothing else.)
2. Then output the thinking and medical reasoning process in
    <think>...</think> tags.
3. Finally, provide the correct answer inside <answer>...</answer>
    tags.
4. Do not include any extra information or text outside of these
    tags.
Question:
<image>{{ content | trim }}
```

---

## A.2 EVALUATION BASE TEMPLATE (SHORT-FORM QA/MCQ)

This judge prompt yields a binary score (0/1) for short-form QA and MCQ-style tasks. It compares the predicted `<answer>` against the reference, allowing paraphrases and option-label matches. Inference is performed with a separate LLM-as-judge (served via vLLM) to reduce evaluation–training coupling. We use deterministic settings (e.g., temperature 0) for reproducibility and parse the returned JSON strictly.

---

**Evaluation BASE template Prompt**

```
You are a medical expert.

Your task is to evaluate whether the Predicted Answer correctly
    answers the Medical Question, based on the Ground Truth
    (Correct Answer) provided.

Question:
{question}

Correct Answer:
{correct_answer}
```

---

```
Predicted Answer:
{predicted_answer}

Score 1 if the predicted answer matches the correct answer either
    fully in text or by indicating the correct option label (e.g.,
    "B", "Option B", or a paraphrased version that clearly
    identifies the correct choice). Score 0 if the predicted answer
    is incorrect or points to the wrong option.

Respond strictly in the following JSON format:

```json
{{
"score": <score>
}}
```
```

### A.3 EVALUATION TEMPLATE FOR REPORT GENERATION

For long-form outputs (e.g., report generation or summarization), the judge assigns a rubric score in [0, 5] reflecting clinical accuracy, completeness, and relevance. We request strict JSON for reliable parsing and average scores across items for dataset-level metrics. Only the model's final report text is provided to the judge; any hidden reasoning (e.g., within </think>) is stripped before evaluation.

**Evaluation Prompt for Report Generation**

```
You are a medical expert evaluating the clinical accuracy,
    completeness, and relevance of a generated medical report or
    summary.

Your task is to compare an AI-generated report or summary to a
    reference (gold standard) report or summary, based on a
    clinical instruction or question. Assess the generated output
    on how well it preserves key clinical information, factual
    correctness, and clinical reasoning relevant to the task.

Assign a score between 0 and 5 using the following scale:

0 - Completely incorrect: Clinically irrelevant, misleading, or
    factually wrong. No meaningful alignment with the instruction
    or reference.

1 - Poor match: Barely relevant or mostly incorrect. Contains
    significant clinical misinformation or omits nearly all
    critical details.

2 - Weak match: Some fragments of relevant content are present, but
    major clinical errors or omissions exist. Clinical utility is
    low.

3 - Fair match: Contains several relevant points, but includes
    notable errors, missing findings, or misinterpretations that
    affect clinical reliability.

4 - Good match: Mostly accurate and clinically sound. Minor issues
    or missing details, but the overall meaning and purpose are
    preserved.
```

```
5 - Perfect or near-perfect match: Clinically accurate, complete,
    and faithful to the instruction and reference. No significant
    omissions or errors.

Respond only in the following example JSON format:

Example JSON format:
```json
{{
"score": <score between 0 and 5>
}}
```

Now, evaluate the following:

### Clinical Instruction or Question::
{question}

### Reference Report or Summary:
{correct_answer}

### AI-Generated Report or Summary:
{predicted_answer}
```

### A.4 HUMAN EXPERT COMPARATIVE EVALUATION PROTOCOL

For a sampled set of multimodal questions, four anonymized model outputs (A–D) plus a reference description are shown; experts pick the single best response based on clinical correctness, relevance (no hallucinations), and clarity of reasoning. Votes are aggregated into preference percentages reported in the main text.

**Evaluation Protocol for Medical Experts**

```
Instructions for Evaluation
Your task is to evaluate the responses provided by three AI models
    based on a given medical image description (Ground Truth).
    Follow these steps to make your selection:

1) Read the Ground Truth: Carefully review the provided description
    of the medical image. This serves as the reference for an
    accurate and detailed response.

2) Assess the Model Responses: Examine the three model-generated
    responses (Model A, Model B, and Model C, Model D). Compare
    their content with the Ground Truth, focusing on the accuracy,
    completeness, and relevance of the clinical reasoning

3) Select the Best Response: Choose the model response that best
    aligns with the Ground Truth in terms of:

 > Clinical Accuracy: Does the response correctly describe the key
    findings in the image?

 > Reasoning Traces: Does the models reasoning traces correct and
    well explained

4) Submit Your Choice: After evaluating the responses, select the
    one that provides the most accurate and comprehensive
    explanation.
```

## A.5 QUALITATIVE EXAMPLES FOR REPORT GENERATION TASK

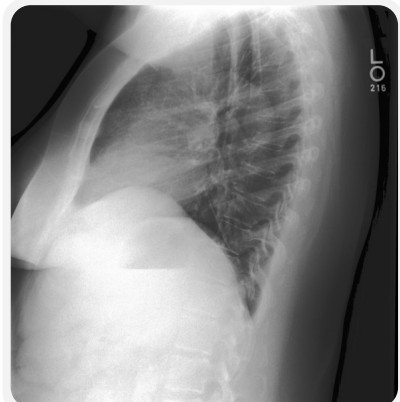

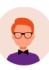

**Generate a detailed report based on the scan**

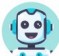

*think*

The image appears to be a chest radiograph (X-ray), likely taken in a posterior-anterior (PA) view, as suggested by the positioning of the heart and spine. The patient seems to be upright during the examination, with the diaphragm in a normal position. The bony structures, including the ribs, clavicles, scapulae, and vertebrae, are clearly visible and show no evidence of fractures, dislocations, or deformities. The lung fields appear clear, with no signs of consolidation, masses, or abnormal opacities. The lung markings are normal, and there is no visible pleural effusion or pneumothorax. The heart size is within normal limits, and the mediastinum appears central without widening or abnormal contours. The diaphragm and costophrenic angles are sharp and well-defined, indicating normal lung expansion and no fluid accumulation

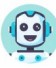

The image provided appears to be a radiograph (X-ray) of the chest. Here is a detailed analysis based on the visible features:

**General Observations:**
- The image is a posterior-anterior (PA) view of the chest, as indicated by the orientation of the spine and the position of the heart shadow.
- The patient is likely standing or sitting upright during the scan, as the diaphragm appears to be in a normal position.

**Key Features:**
**1. Bony Structures:**
  - The ribs are visible on both sides, and the clavicles are also visible.
  - The spine is well-defined, with the vertebrae appearing intact and without any obvious fractures or dislocations.
  - The scapulae are visible on both sides, and the shoulder joints appear to be in a normal position.

**2. Lungs:**
  - The lung fields appear to be clear without any visible opacities, masses, or infiltrates.
  - The lung markings are visible and appear to be within normal limits.
  - There is no evidence of pleural effusion or pneumothorax.

**3. Heart and Mediastinum:**
  - The heart size appears to be within normal limits.
  - The mediastinum is not widened, and there are no visible masses or abnormalities in this region.

**4. Diaphragm:**
  - The diaphragm appears to be intact and in a normal position.
  - The costophrenic angles (where the diaphragm meets the ribs) appear sharp and clear, indicating no pleural effusion.

**Potential Concerns:**
- There are no obvious fractures or dislocations in the bony structures.
- The lung fields appear clear, and there are no visible masses or infiltrates.
- The heart size and mediastinum appear normal.
- The diaphragm is intact and in a normal position.

**Conclusion:**
Based on the radiographic findings, this chest X-ray appears to be within normal limits. There are no immediate concerns noted in this image. However, a definitive interpretation should be made by a qualified radiologist who can consider the patient's clinical history and symptoms.

If there are any specific symptoms or concerns, it is recommended to consult a healthcare provider for further evaluation.

Figure 5: **MediX-R1 - Report Generation: Case 1**

## A.6 HUMAN EVALUATION: MODEL REASONING

We extend our human expert study detailed in (Sec. 4.3) to evaluate the reasoning quality of our MediX-R1 model against MedGemma with the help of medical doctors. Experts assessed outputs for clinical accuracy, reasoning soundness, and practical usefulness in a medical setting. MediX-R1's reasoning was preferred in 74.2% of cases over MedGemma, indicating stronger clinical coherence. Furthermore, the study shows that in 92.4% of the cases, the model's reasoning steps were rated as acceptable and often comparable to a medical doctor's thought process, while only 7.6% of the cases were rated as having poor reasoning quality. Moreover, in fewer than 5% of the cases,

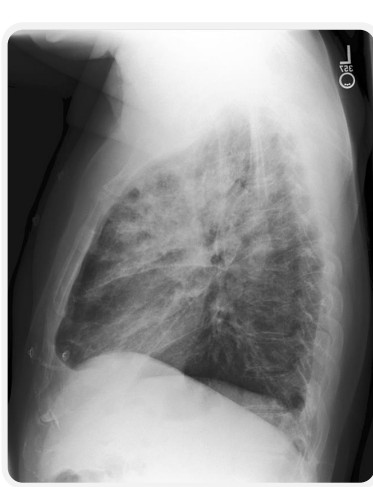

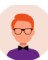

**Generate a detailed report based on the scan**

*think*

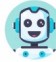

The image is a radiograph, specifically a chest X-ray, which is used to assess the lungs, heart, and mediastinum. The view appears to be posteroanterior (PA), as indicated by the relative heart size and orientation of the structures. The diaphragm is in a normal position, suggesting the patient was upright during the scan.

On inspection, there is increased opacification noted in the right lung field, which may represent an abnormal finding. This could be due to several potential causes such as infection (pneumonia), pulmonary edema, mass lesion, or other infiltrative processes. The left lung field appears relatively clear in comparison. The heart size and mediastinum are within normal limits, showing no evidence of enlargement or shift. The diaphragm is intact with clear costophrenic angles, indicating no pleural effusion.

The bony structures, including the ribs, clavicles, and scapulae, are visible and intact without evidence of fracture or deformity. Overall, the primary observation is the increased opacity in the right lung, which warrants further clinical correlation and radiological evaluation to determine the underlying cause. A detailed review by a qualified radiologist is recommended for confirmation and diagnosis.

The image provided is a chest X-ray. Here is a detailed report based on the visible findings:

General Observations:
- The X-ray is a posteroanterior (PA) view, as indicated by the orientation of the heart and the position of the diaphragm.
- The patient's left side is on the right side of the image, and the right side is on the left side of the image, which is typical for a PA view.

Lung Fields:
- The lung fields appear to have increased opacification, particularly in the right lung field. This increased opacification could be due to several reasons, including but not limited to:
  - Infiltration (e.g., pneumonia, pulmonary edema, or consolidation).
  - Mass lesion.
  - Pleural effusion.
  - Other pathologies such as fibrosis or tumor.
- The left lung field appears relatively clear compared to the right lung field.

Heart:
- The heart size appears to be within normal limits on this view.
- The mediastinum appears to be within normal limits.

Diaphragm:
- The diaphragm appears to be intact and without any obvious abnormalities.
- The costophrenic angles appear clear, which is a normal finding.

Bones:
- The bony structures of the thorax, including the ribs, clavicles, and scapulae, appear intact without any obvious fractures or lytic lesions.

Pleural Spaces:
- There appears to be no evidence of pleural effusion on this view.

Summary:
The X-ray shows increased opacification in the right lung field, which could be indicative of a variety of conditions. Further clinical correlation with the patient's history, symptoms, and other diagnostic tests (such as a CT scan or sputum analysis) would be necessary to determine the exact cause of the opacification. A radiologist's interpretation is recommended for a definitive diagnosis.

Recommendations:
- The patient should be evaluated by a healthcare provider for further investigation and management.
- Additional imaging or tests may be required to determine the cause of the opacification in the right lung field.

This report is for informational purposes only and should not be considered a substitute for a professional medical opinion.

Figure 6: **MediX-R1 - Report Generation: Case 2**

the model produced flawed reasoning despite generating the correct final answer, indicating that such inconsistencies are rare and that MediX-R1 generally maintains a robust and coherent reasoning process. Reviewers comprised five certified medical experts (MBBS/MD) with specialties in Radiology, General Medicine, and Forensic Medicine, with an inter-rater agreement of 63%.

## A.7 TRAINING DATA AND MODALITY DISTRIBUTION

We trained MediX-R1 on 51335 multimodal medical instruction samples spanning 16 modality tags. All samples were drawn from the official train splits of the source datasets: PMC-VQA subset (Zhang et al., 2024), SLAKE (Liu et al., 2021), RadVQA (Lau et al., 2018), and PathVQA (He et al., 2020).

| Medical Modality | Samples |
|---|---|
| X_RAY | 5964 |
| MICROSCOPY | 16399 |
| CLINICAL_PHOTOGRAPHY | 8979 |
| CT_SCAN | 7646 |
| GRAPHICS | 2205 |
| ANGIOGRAPHY | 522 |
| PET_SCAN | 406 |
| ULTRASOUND | 1227 |
| MRI_SCAN | 6224 |
| FUNDUS_PHOTOGRAPHY | 314 |
| OCT_SCAN | 236 |
| ENDOSCOPY | 611 |
| MAMMOGRAPHY | 106 |
| FLUOROSCOPY | 321 |
| OTHER | 64 |
| SPECT | 111 |
| **Total** | **51335** |

| Dataset | Samples |
|---|---|
| PMC_VQA_SUBSET | 25000 |
| SLAKE | 4919 |
| RAD_VQA | 1793 |
| PATH | 19623 |
| **Total** | **51335** |

Table 5: **Modality Breakdown and Source Dataset composition**

## A.8 TRAINING CONFIGURATION

We list below the GRPO training configuration used for MediX-R1. Core settings include (i) data filtering and batching, (ii) actor optimization and rollout sampling, (iii) KL-regularized GRPO advantage computation, and (iv) trainer settings. We train our models using the EasyR1(Zheng et al., 2025b) Github Repository. MediX-R1 was trained using 8×A100 (80 GB) Nvidia GPUs for approximately 25 hours.

**Training Configuration**

```
Training Configurations
  "data": {
    "max_prompt_length": 4352,
    "max_response_length": 4096,
    "rollout_batch_size": 512,
    "val_batch_size": 1024,
    "shuffle": true,
    "seed": 1,
    "min_pixels": 262144,
    "max_pixels": 4194304,
    "filter_overlong_prompts": true,
    "filter_overlong_prompts_workers": 16
  },
  "worker": {
    "hybrid_engine": true,
    "actor": {
      "strategy": "fsdp",
      "global_batch_size": 128,
      "micro_batch_size_per_device_for_update": 1,
```

```
          "micro_batch_size_per_device_for_experience": 2,
          "max_grad_norm": 1.0,
          "clip_ratio_low": 0.2,
          "clip_ratio_high": 0.3,
          "clip_ratio_dual": 3.0,
          "loss_avg_mode": "token",
          "padding_free": true,
          "dynamic_batching": true,
          "use_torch_compile": true,
          "optim": {
            "lr": 1e-6,
            "betas": [0.9, 0.999],
            "weight_decay": 0.01,
            "strategy": "adamw",
            "lr_scheduler_type": "constant",
            "training_steps": 200
          },
          "fsdp": {
            "enable_full_shard": true,
            "enable_rank0_init": true,
            "mp_param_dtype": "bf16",
            "mp_reduce_dtype": "fp32",
            "mp_buffer_dtype": "fp32"
          },
          "offload": {
            "offload_params": true,
            "offload_optimizer": true
          },
          "use_kl_loss": true,
          "kl_penalty": "low_var_kl",
          "kl_coef": 0.01
        },
        "rollout": {
          "name": "vllm",
          "n": 5,
          "temperature": 1.0,
          "top_p": 1.0,
          "seed": 1,
          "tensor_parallel_size": 2,
          "max_num_batched_tokens": 8448,
          "gpu_memory_utilization": 0.6,
          "val_override_config": {
            "temperature": 0.6,
            "top_p": 0.95,
            "n": 1
          },
          "prompt_length": 4352,
          "response_length": 4096
        }
      },
      "algorithm": {
        "adv_estimator": "grpo",
        "gamma": 1.0,
        "lam": 1.0,
        "use_kl_loss": true,
        "kl_penalty": "low_var_kl",
        "kl_coef": 0.01,
        "kl_type": "fixed",
        "kl_target": 0.1,
        "kl_horizon": 10000.0
      },
      "trainer": {
```

```
    "total_epochs": 2,
    "nnodes": 1,
    "n_gpus_per_node": 8,
    "val_freq": 5,
    "val_before_train": true,
    "save_freq": 5,
    "save_limit": 3
}
```

## A.9 REWARD FUNCTION SOURCE CODE

Below are the Python implementations of the four reward components used in MediX-R1. Each function operates on a predicted model output string and a ground truth string containing the modality tag and reference answer.

**Format reward**

```python
def format_reward(predict: str) -> float:
    idx = predict.find("<think>")
    if idx == -1:
        return 0.0
    predict_new = predict[idx:].strip()
    pattern = re.compile(r"<think>.*?</think>\s*<answer>.*?</answer>"
        , re.DOTALL)
    format_match = re.fullmatch(pattern, predict_new)
    return 1.0 if format_match else 0.0
```

**LLM-based accuracy reward**

```python
def accuracy_reward_llm(predict: str, ground_truth: str) -> float:
    try:
        content_match = re.search(r"<answer>(.*?)</answer>", predict,
            re.DOTALL)
        given_answer = content_match.group(1).strip() if content_match
            else predict.strip()
        given_answer = given_answer.strip('.')
        ground_truth = ground_truth.split('>', maxsplit=1)[1].strip()
        ground_truth = ground_truth.strip('.')

        if given_answer == '' or len(given_answer) == 1:
            return 0.0
        if given_answer == ground_truth:
            return 1.0
        llm_score = llm_answer_match(given_answer, ground_truth) #
            external helper
        return llm_score
    except Exception:
        return 0.0
```

**Embedding-based semantic reward**

```python
def accuracy_reward_embed(predict: str, ground_truth: str, threshold
    : float = 0.8) -> float:
    try:
```

```
        content_match = re.search(r"<answer>(.*?)</answer>", predict,
            re.DOTALL)
        given_answer = content_match.group(1).strip() if content_match
            else predict.strip()
        given_answer = given_answer.strip('.')
        ground_truth = ground_truth.split('>', maxsplit=1)[1].strip()
        ground_truth = ground_truth.strip('.')

        if given_answer == '' or len(given_answer) == 1:
            return 0.0
        if given_answer == ground_truth:
            return 1.0

        embeddings = embed_model.encode([given_answer, ground_truth],
            convert_to_tensor=True)
        similarity = util.pytorch_cos_sim(embeddings[0], embeddings
            [1]).item()
        return float(similarity >= threshold)
    except Exception:
        return 0.0
```

**Modality recognition reward**

```
def modality_reward(predict: str, ground_truth: str) -> float:
    idx = predict.find("<think>")
    if idx == -1:
        return 0.0
    predict_new = predict[:idx].strip() # modality tag before <think>
    modality = ground_truth.split('>', maxsplit=1)[0] + '>'
    return 1.0 if predict_new.upper() == modality.upper() else 0.0
```

## A.10   EVALUATION ON REAL WORLD CLINICAL DATA

To further assess the generalization ability of our model, we conducted additional evaluation on MedPix 2.0 (Siragusa et al., 2025), a publicly available real-world clinical VQA dataset derived from the original MedPix (Henigman & Kennedy, 2025) database maintained by the U.S. National Library of Medicine (NIH). MedPix comprises over 12,000 anonymized, crowdsourced clinical cases containing medical images and corresponding textual information such as findings, diagnoses, and treatments. This ensures both reproducibility and compliance with NIH privacy standards.

The evaluation on MedPix 2.0 demonstrates that our model, MediX-R1, consistently outperforms other medical vision-language models. Specifically, MediX-R1 achieves a score of 51.11%, surpassing strong baselines and previous SOTA Medical Models as shown in Table 6. These results further confirm the robustness and adaptability of MediX-R1 on diverse real-world clinical data, emphasizing its capability to generalize beyond controlled experimental environments.

| Model | Score (%) |
|---|---|
| MedVLM-R1 | 27.57 |
| MedGemma | 43.18 |
| LLaVA-Med | 44.29 |
| BiMediX2 | 46.51 |
| HuatuoGPT | 48.81 |
| **MediX-R1 (Ours)** | **51.11** |

Table 6: Performance comparison on the MedPix 2.0 dataset.

