# OpenReview forum: "MediX-R1:  Open Ended Medical Reinforcement Learning"
_ICLR.cc/2026/Conference — Submitted to ICLR 2026_

### Official Review · Reviewer_jMAA · 2025-10-30

**Soundness:** 2
**Presentation:** 3
**Contribution:** 2
**Rating:** 2
**Confidence:** 4

**Summary:**

Authors present MediX-R1, a multimodal RLVR framework for clinical applications, which uses GRPO with four complementary rewards: (i) an LLM-as-a-judge accuracy check, (ii) an embedding-based semantic similarity reward, (iii) a format reward for special tokens, and (iv) a modality recognition reward tfor cross-modality hallucination. According to authors claim, trained on ~50K medical instructions, MediX-R1 achieves better results across both text-only and image+text medical tasks under proposed three-stage LLM-as-judge evaluation pipeline. Authors also investigate reward-hacking failure modes and present human qualitative analyses as part of their ablation studies.

**Strengths:**

- Authors focus on a timely topic on RLVR-based medical concepts, especially given the recent rise of RLVR approaches in text-only reasoning tasks.
- Authors provide a structured reward design consisting of four complementary components, which can be useful for future research in medical RLVR and multimodal-RL.

**Weaknesses:**

- Proposed method appears to be a straightforward application of existing GRPO algorithm with multimodal medical data, without new algorithmic improvements. Some popular open-source RL frameworks like Verl and Easy-R1 already support multimodal training and the core work seems to be primarily reward choices rather than theoretical contributions. Moreover, eventhough authors claim to provide an open-source framework as contribution, there is no code provided during submission which makes it impossible for reviewers to verify the implementation or validate this stated contribution.
- One of the major weakness of the paper is too much reliability on LLM-as-a-Judges both in reward design and evaluation. Using LLM judges requires a detailed reliability analysis such as multiple runs, human correlations, and testing with different LLMs, in which authors conduct none of these; furthermore, the LLM judge used is a small Qwen3-4B in verifier, which further reduces reliability. On the other hand, authors evaluate their framework with an Qwen-14B model only, which again lacks the aforementioned issues. Although the authors claim to have qualitatively examined outputs with human experts, the provided human evaluation is very narrow and weak; it should demonstrate human correlation with a sufficient number of expert annotators with more details like their experience level and the professional focuses (e.g., X% portion radiology). Thus, their LLM-as-judge evaluation lacks reliability for considering the results credible.
- Provided ablation studies and comparisons are not comprehensive enough: authors does not compare with other RL methods (PPO, DAPO, REINFORCE++, etc.) to justify the choice of GRPO and there are no experiments with different base models (e.g., Llama) or different model scales (smaller/larger Qwen models) to demonstrate generalization.
- Finally, authors mention they use ~50K training data to RL-finetune MediX-R1 as a contribution in the abstract and at the end of the introduction, but the details of data construction and training dynamics are not discussed.

**Questions:**

1. Could authors share the human experiment details like the number of expert, their backgrounds (radiologists, generalists?), inter-annotator agreement, and annotation protocol?
2. How is the training data constructed and what training dynamics are followed (i.e., training setting details)?

---

> ### Author Response · Authors · 2025-12-01
>
> Thank you for your thoughtful review and for recognizing the relevance and structure of our work. We appreciate your positive feedback on our focus on RLVR-based medical reasoning and our proposed multi-component reward framework. We are encouraged that you find the structured reward design valuable for advancing future research in multimodal and medical reinforcement learning.
>
> ---
>
> ## ***W1. Open sourcing code and models & Open-Ended Medical Reinforcement Learning Contributions***
>
> Thank you for your suggestion, we have added the python code for our composite reward system in the Appendix Sec A.9 to ensure reproducibility. We commit to releasing our model weights, full code base with training and evaluation framework, curated datasets will be released on open source platforms like Huggingface and Github.
>
> While our work builds upon the GRPO RL framework, we found that standard GRPO formulations are not directly applicable to the medical domain, where open-ended, clinically grounded responses are required. Existing RL/GRPO-trained models (e.g., DeepSeek-R1[1], MedVLM-R1[2]) often rely on deterministic rewards such as MCQs or short-form answers. However, practical medical responses rarely fit such rigid formats, clinical explanations, symptom descriptions, or diagnostic reasoning can be expressed in diverse yet equally valid ways, even among experts. This makes simple string-matching or single character correctness metrics insufficient and often misleading in evaluating medical responses.
>
> To validate the effectiveness of our method,  we conducted additional experiments using the default GRPO configuration (i.e., format + accuracy reward with simplified string matching). We found that this default setting largely optimized for MCQ style or short answer evaluation fails to generalize to open-ended medical responses. Specifically, the constrained reward formulation biases the model toward rigid answer structures, leading to degraded performance on tasks requiring flexible clinical reasoning. While MCQ and short answer accuracies remained consistent or even improved under default GRPO training/reward (showing a 3.22% overall increase over baseline), open ended QA performance deteriorated sharply, with long form response accuracy dropping by 33.86%. This empirical result further highlights the necessity of our proposed reward design for reliably training models that can perform well across both structured and unstructured clinical tasks.
>
> To address this, we extend GRPO to handle open-ended medical responses with limited training data and introduce a composite reward system that integrates correctness, semantics, modality grounding, and format compliance. This richer reward signal helps mitigate reward hacking and stabilizes training, a common challenge in RL frameworks. We believe these design and empirical insights are valuable contributions that can generalize to other domains requiring open-ended reasoning.
> Furthermore, we introduce a unified three-stage evaluation framework combining LLM (text-only) and VLM (image+text) assessment to benchmark medical multimodal reasoning models more comprehensively.
>
> *[1] Guo, Daya, et al. "Deepseek-r1: Incentivizing reasoning capability in llms via reinforcement learning." arXiv preprint arXiv:2501.12948 (2025).*
>
> *[2] Pan, Jiazhen, et al. "Medvlm-r1: Incentivizing medical reasoning capability of vision-language models (vlms) via reinforcement learning." International Conference on Medical Image Computing and Computer-Assisted Intervention. Cham: Springer Nature Switzerland, 2025.*

---

> > ### Author Response · Authors · 2025-12-01
> >
> > ---
> >
> > ## ***W2. On LLM-as-a-Judge***
> >
> > To ensure robustness, we perform controlled evaluations using deterministic generation settings (temperature = 0, top_p = 1) and averaged results over three runs, observing only ±0.002 variation. For additional validation, we replaced Qwen3-14B with GPT-5.1 and GPT-5 mini as evaluators, which resulted in a deviation of only ±0.005, indicating high consistency across models.
> >
> > In both our training and evaluation pipeline the task of the LLM is to only assign a binary value (yes/no) if the predicted response matches with the ground truth response, and LLMs have shown to be superior for this task compared to traditional string overlap metrics like (BLEU,ROUGE,F1) Also acknowledged by reviewer Z9Lo. We choose a light weight model like Qwen3-4B in our training pipeline to optimize for training compute as our approach requires hosting the evaluator model during the training phase.
> >
> > We have conducted our human expert study with the help of 5 certified medical experts with MBBS or MD degrees in Medicine with specialities including - Radiology, General Medicine, and Forensics. We observe an inter-rater agreement of 63% among these experts. Two experts are also the authors of this paper and have been involved in the diagnosis and design of the entire research phase. Additional details about the qualification and experience of the doctors will be added to the final draft to preserve the anonymity of the review process. Details regarding the reasoning evaluation and evaluation protocol is added in Sec A.4 and A.6.

---

> ### Author Response · Authors · 2025-12-01
>
> ## ***W3. On different RL Algorithms and backbones***
>
> We appreciate the reviewer’s valuable suggestion to expand  the scope of our ablation studies to additional RL approaches and backbones. As suggested, we have now performed additional experiments as described below:
>
> At first, we conducted additional experiments using RL methods beyond GRPO.  We observe that our composite reward is effective across multiple RL methods, including DAPO and GSPO. Under the same setting and only 1 epoch training, DAPO achieved a score of 60.72%, GRPO attained 59.61%, and GSPO achieved 59.69%, all of them outperform the baseline Qwen2.5-VL (57%) across MCQ based, short answer QA, and  and longform open ended QA.  These results demonstrate that the proposed composite reward model generalizes well across RL approaches.
>
> Second, we trained our method on different backbones, the latest Qwen3-VL and observed a ~2% gain by our method even on this stronger backbone. In addition, we experimented with our composite rewards on SmolVLM2 [1] backbone, observing a 2.2% gain. These consistent gains across diverse model families demonstrate that MediX-R1 enhances reasoning ability beyond model-specific reward tuning.
>
> Together, these results show that our findings are not specific to GRPO and that the composite reward model delivers consistent gains across RL algorithms and model backbones.
>
> *[1] Marafioti, Andrés, et al. "Smolvlm: Redefining small and efficient multimodal models." arXiv preprint arXiv:2504.05299 (2025).*
>
> ---
>
> ## ***W4. On Training Data Source***
>
> We apologize for the oversight. The training data was sourced from existing open source medical datasets given below.
>
> The detailed breakdown of our training instructions along with their respective modality is given below. Here, the training samples were prepared by filtering out extremely small images with image height or width less than 28 pixels.
>
> **Training Data:** Total samples: 51,335
>
> **Medical Modality breakdown**
>
> | **Medical Modality** | **Samples #** |
> |:--------------------|--------------:|
> | X_RAY                |         5,964 |
> | MICROSCOPY           |        16,399 |
> | CLINICAL_PHOTOGRAPHY |         8,979 |
> | CT_SCAN              |         7,646 |
> | GRAPHICS             |         2,205 |
> | ANGIOGRAPHY          |           522 |
> | PET_SCAN             |           406 |
> | ULTRASOUND           |         1,227 |
> | MRI_SCAN             |         6,224 |
> | FUNDUS_PHOTOGRAPHY   |           314 |
> | OCT_SCAN             |           236 |
> | ENDOSCOPY            |           611 |
> | MAMMOGRAPHY          |           106 |
> | FLUOROSCOPY          |           321 |
> | OTHER                |            64 |
> | SPECT                |           111 |
> | **Total**            |    **51,335** |
>
> **Training Dataset Sources**
>
> | **Dataset**    | **Samples #** |
> |----------------|--------------:|
> | PMC_VQA_SUBSET |         25000 |
> | SLAKE          |          4919 |
> | RAD_VQA        |          1793 |
> | PATH           |         19623 |
> | **Total**      |     **51335** |
>
> Our training dataset has been curated from the corresponding train split of the source dataset.
>
> We have added these tables to the Appendix Sec A.7. The training configuration is added to Appendix Sec A.8.
>
> ---
>
> ## ***Q1. On Human Expert Evaluation***
>
> Thank you for your comment. We have conducted our human expert study with the help of 5 certified medical experts with MBBS or MD degrees in Medicine with specialities including - Radiology, General Medicine, and Forensics. We observe an inter-rater agreement of 63% among these experts. Two experts are also the authors of this paper and have been involved in the diagnosis and design of the entire research phase. Additional details about the qualification and experience of the doctors will be added to the final draft to preserve the anonymity of the review process. Details regarding the evaluation protocol are added in Sec A.4.
>
> Additionally, we have extended our human expert study to evaluate the reasoning quality of our MediX-R1 model against MedGemma with the help of medical doctors as detailed in (Sec A.6). Experts assessed outputs for clinical accuracy, reasoning soundness, and practical usefulness in a medical setting. MediX-R1’s reasoning was preferred in 74.2% of cases over MedGemma, indicating stronger clinical coherence. Furthermore,  the study shows that in 92.4% of the cases, the model’s reasoning steps were rated as acceptable and often comparable to a medical doctor’s thought process, while only 7.6% of the cases were rated as having poor reasoning quality. Moreover, in fewer than 5% of the cases, the model produced flawed reasoning despite generating the correct final answer, indicating that such inconsistencies are rare and that MediX-R1 generally maintains a robust and coherent reasoning process. These details are updated in Sec A.6
>
> ---
>
> ## ***Q2. On training dataset details***
>
> Please refer to the response provided in W4, which explains this in detail.

---

### Official Review · Reviewer_MGH6 · 2025-10-31

**Soundness:** 2
**Presentation:** 3
**Contribution:** 2
**Rating:** 6
**Confidence:** 4

**Summary:**

The paper presents MediX-R1, a framework for open-ended reinforcement learning (RL) in medical multimodal language models. The key idea is to train a vision-language model (Qwen2.5-VL 7B) using Group Relative Policy Optimization (GRPO) and a composite reward tailored for clinical reasoning. A three-stage LLM-as-judge evaluation pipeline assesses both text-only and multimodal outputs. The proposed MediX-R1 achieves improvements across 18 medical benchmarks, outperforming strong baselines such as BiMediX2, MedGemma, and HuatuoGPT-V.

**Strengths:**

- **Novel reward design**: The multi-signal reward (LLM + embeddings + modality + format) is well-motivated and shown to stabilize open-ended RL.
- **Unified evaluation**: The proposed LLM-as-judge framework provides a consistent and semantically aware assessment across text and image tasks.
- **Broad modality coverage**: Supports diverse imaging types (CT, X-ray, MRI, microscopy, ultrasound, etc.), improving the model’s clinical versatility.

**Weaknesses:**

- **High dependence on pretrained Qwen2.5-VL**: The improvements might reflect reward tuning rather than genuine reasoning advancement; no results are shown on alternative backbones.
- **Limited transparency in human evaluation**: The human evaluation section lacks important details—there is no information about the number of annotators, their medical background, or the inter-rater agreement between them. Additional information would increase the reliability of the results.
- **Limited novelty in RL design**: The framework primarily combines existing techniques without clear new algorithmic steps, although one of the contributions is "introducing open-ended medical reinforcement learning".

**Questions:**

1. What is the total compute budget and training time for MediX-R1 compared to multi-stage baselines like BiMediX2 or MedGemma?
2. How well does MediX-R1 generalize to unseen modalities or real-world hospital data not represented in training sets?
3. Have the authors tested or considered evaluating MediX-R1 on alternative backbones (e.g., LLaVA-Med) to demonstrate broader generalization?

---

> ### Author Response · Authors · 2025-12-01
>
> Thank you for your thoughtful and encouraging review. We sincerely appreciate your recognition of our contributions, particularly the novel multi-signal reward design, the unified LLM-as-judge evaluation framework, and the broad modality coverage of MediX-R1. Your positive feedback reinforces our motivation to further refine the framework and continue advancing open-ended RL for multimodal clinical reasoning.
>
> ---
>
> ## ***W1. Results on alternative backbones***
>
> We thank the reviewer for the suggestion to extend the experiments beyond the Qwen2.5-VL backbone. As suggested, to assess generality, we also trained our model using a different backbone; the latest Qwen3-VL and observed a ~2% gain by our method even on this stronger backbone. In addition, we experimented with our composite rewards on SmolVLM2 [1] backbone, observing a 2.2% gain. These consistent gains across diverse model families demonstrate that MediX-R1 enhances reasoning ability beyond model-specific reward tuning.
>
> *[1] Marafioti, Andrés, et al. "Smolvlm: Redefining small and efficient multimodal models." arXiv preprint arXiv:2504.05299 (2025).*
>
> ---
>
> ## ***W2. On Human Expert Evaluation***
>
> Thank you for your comment. We have conducted our human expert study with the help of 5 certified medical experts with MBBS or MD degrees in Medicine with specialities including - Radiology, General Medicine, and Forensics. We observe an inter-rater agreement of 63% among these experts. Two experts are also the authors of this paper and have been involved in the diagnosis and design of the entire research phase. Additional details about the qualification and experience of the doctors will be added to the final draft to preserve the anonymity of the review process.
>
> Additionally, we have extended our human expert study to evaluate the reasoning quality of our MediX-R1 model against MedGemma with the help of medical doctors. Experts assessed outputs for clinical accuracy, reasoning soundness, and practical usefulness in a medical setting. MediX-R1’s reasoning was preferred in 74.2% of cases over MedGemma, indicating stronger clinical coherence. Furthermore,  the study shows that in 92.4% of the cases, the model’s reasoning steps were rated as acceptable and often comparable to a medical doctor’s thought process, while only 7.6% of the cases were rated as having poor reasoning quality. Moreover, in fewer than 5% of the cases, the model produced flawed reasoning despite generating the correct final answer, indicating that such inconsistencies are rare and that MediX-R1 generally maintains a robust and coherent reasoning process. These details are updated in Sec A.6

---

> > ### Author Response · Authors · 2025-12-01
> >
> > ---
> >
> > ## ***W3. Open-Ended Medical Reinforcement Learning Contributions***
> >
> > Thank you for your thoughtful comment. While our work builds upon the GRPO RL framework, we found that standard GRPO formulations are not directly applicable to the medical domain, where open-ended, clinically grounded responses are required. Existing RL/GRPO-trained models (e.g., DeepSeek-R1 [1], MedVLM-R1 [2]) typically rely on deterministic reward structures such as MCQs or short-form answers. However, practical medical responses rarely fit such rigid formats; clinical explanations, symptom descriptions, or diagnostic reasoning can be expressed in diverse yet equally valid ways, even among experts. This makes simple string-matching or single-character correctness metrics insufficient and often misleading for evaluating medical responses.
> >
> > To address this, we extend GRPO to better handle open-ended medical responses under limited training data. We introduce a composite reward system that integrates correctness, semantics, modality grounding, and format compliance. This richer, multi-dimensional reward signal helps reduce reward hacking and stabilizes training issues commonly observed in RL frameworks when the reward structure is overly rigid or sparse. We believe these design and empirical insights constitute meaningful contributions that can generalize to other domains requiring open-ended reasoning.
> >
> > To further validate the effectiveness of our method, we conducted additional experiments using the default GRPO configuration (i.e., format + accuracy reward with simplified string matching). We found that this default setting largely optimized for MCQ style or short answer evaluation fails to generalize to open-ended medical responses. Specifically, the constrained reward formulation biases the model toward rigid answer structures, leading to degraded performance on tasks requiring flexible clinical reasoning. While MCQ and short answer accuracies remained consistent or even improved under default GRPO training/reward (showing a 3.22% overall increase over baseline), open ended QA performance deteriorated sharply, with long form response accuracy dropping by 33.86%. This empirical result further highlights the necessity of our proposed reward design for reliably training models that can perform well across both structured and unstructured clinical tasks.
> >
> > Furthermore, we conducted additional experiments using alternative RL frameworks. Our composite reward is effective across multiple RL methods, including DAPO and GSPO. Under the same setting and 1 epoch training, DAPO achieved a score of 60.72%, GRPO attained 59.61%, and GSPO achieved 59.69%, all of them are above the baseline Qwen2.5-VL (57%) ) across diverse MCQ based, short answer QA, and  and longform open ended QA types. These results demonstrate that the proposed composite reward model generalizes well across RL approaches.
> >
> > Finally, we introduce a unified three-stage evaluation framework combining LLM-based (text-only) and VLM-based (image+text) assessment to more comprehensively benchmark medical multimodal reasoning. This provides a more realistic and holistic evaluation of open-ended clinical performance than existing metrics.
> >
> > *[1] Guo, Daya, et al. "Deepseek-r1: Incentivizing reasoning capability in llms via reinforcement learning." arXiv preprint arXiv:2501.12948 (2025).*
> >
> > *[2] Pan, Jiazhen, et al. "Medvlm-r1: Incentivizing medical reasoning capability of vision-language models (vlms) via reinforcement learning." International Conference on Medical Image Computing and Computer-Assisted Intervention. Cham: Springer Nature Switzerland, 2025.*
> >
> > ---
> >
> > ## ***Q1. On Training Compute***
> >
> > MediX-R1 was trained using a single-stage RL framework on 8×A100 (80 GB) GPUs for approximately 25 hours. In comparison, BiMediX2[1] employs a multi-stage training pipeline using 8×AMD Instinct MI200 (64 GB) GPUs and requires roughly 52 hours of total training. The training configuration and compute budget for MedGemma[2] have not been publicly reported, making a direct comparison infeasible.
> >
> > *[1] Mullappilly, Sahal Shaji, et al. "Bimedix2: Bio-medical expert lmm for diverse medical modalities." arXiv preprint arXiv:2412.07769 (2024).*
> >
> > *[2] Sellergren, Andrew, et al. "Medgemma technical report." arXiv preprint arXiv:2507.05201 (2025).*

---

> ### Author Response · Authors · 2025-12-01
>
> ---
>
> ## ***Q2. Evaluation on real-world Clinical data.***
>
> We appreciate the reviewer’s suggestion regarding evaluation on real-world hospital data. Currently, we have not tested our model with modalities beyond X-ray, CT, MRI, Microscopy/Histopathology, Ultrasound, Fluoroscopy, Endoscopy, Angiography, Mammography, Clinical Photography, SPECT (Single Photon Emission Computed Tomography), OCT (Optical Coherence Tomography), and Fundus imaging. We believe that including even a limited subset of additional modalities would further help the system adapt and learn effectively. While access to proprietary clinical data is limited due to privacy and regulatory constraints, we agree that such evaluation would further strengthen our findings.
>
> To address this concern, we extended our evaluation to the MedPix 2.0 [1] dataset, a publicly available, real world clinical dataset derived from the original MedPix [2] database maintained by the U.S. National Library of Medicine (NIH). MedPix contains over 12,000 authentic crowdsourced clinical cases with medical images and corresponding textual information (findings, diagnoses, treatments, etc.), all fully anonymized under NIH privacy protocols.
>
> Using this real world dataset ensures reproducibility while preserving patient privacy. The evaluation results on MedPix 2.0 shows that our model continues to outperform other medical VLMs. Specifically, MediX-R1 achieves a score of 51.11%, surpassing models such as HuatuoGPT (48.81%), BiMediX2 (46.51%), LLaVA-Med (44.29%), MedGemma (43.18%) and MedVLM-R1 (27.57). This result further confirms the robustness and generalization capability of our approach on real-world medical data. We have updated these results in our Appendix Sec A.10
>
> *[1] Siragusa, I., Contino, S., La Ciura, M., Alicata, R., & Pirrone, R. (2024). Medpix 2.0: a comprehensive multimodal biomedical dataset for advanced AI applications. arXiv preprint arXiv:2407.02994, 16.*
>
> *[2] Henigman, A., & Kennedy, B. (2025). MedPix®: database of medical images, teaching cases, and clinical topics. Medical Reference Services Quarterly, 44(3), 328-333.*
>
> ## ***Q3. Results on alternative backbones***
>
> We selected Qwen 2.5 VL as the primary backbone due to its strong community support and its compatibility with RL-based training pipelines. As suggested, to assess generality, we also trained our model using a different backbone; the latest Qwen3-VL and observed a ~2% gain by our method even on this stronger backbone. In addition, we experimented with our composite rewards on SmolVLM2 [1] backbone, observing a 2.2% gain. These consistent gains across diverse model families demonstrate that MediX-R1 enhances reasoning ability beyond model-specific reward tuning.
>
> *[1] Marafioti, Andrés, et al. "Smolvlm: Redefining small and efficient multimodal models." arXiv preprint arXiv:2504.05299 (2025).*

---

### Official Review · Reviewer_Z9Lo · 2025-11-01

**Soundness:** 3
**Presentation:** 3
**Contribution:** 3
**Rating:** 6
**Confidence:** 3

**Summary:**

This paper presents MediX-R1, an open-ended reinforcement learning framework for medical multimodal reasoning. Its key distinction from previous work lies in its richer, composite reward model, which integrates correctness, semantic similarity, format compliance, and modality grounding. Compared with earlier RL-based medical VLMs, this makes the reward signal more fine-grained and clinically aligned. While some may question the reliability of using an LLM-as-judge for reward evaluation, I find this design both reasonable and effective — it provides a far more flexible and informative signal than rigid string matching in medical question-answering setups. Overall, this is a paper I genuinely enjoyed reading it.

**Strengths:**

- The paper is very well written and clearly structured, making complex reinforcement-learning ideas accessible and easy to follow.

- It presents a thoughtful and well-engineered reward model, combining multiple complementary signals (correctness, semantics, modality, and format) in a coherent way.

- The one-stage training design keeps the framework simple yet effective — a welcome contrast to multi-phase or agentic pipelines that often add unnecessary complexity.

**Weaknesses:**

- Limited technical innovation. The work primarily extends existing GRPO-style reinforcement learning with a richer reward structure rather than proposing a new algorithm. While this may be acceptable given the paper’s focus on design and empirical validation, the technical novelty is relatively modest.

- Reproducibility and openness. The paper promises code and model release but does not yet provide them. Given the number of moving parts in the composite reward setup, open-sourcing both code and model weights would be critical for reproducibility.

- Fixed reward weighting. The choice of static coefficients  appears somewhat arbitrary. These hand-tuned weights may strongly influence outcomes. It would strengthen the work to include ablation or sensitivity analyses on these weights, or to explore making them learnable parameters through meta-optimization or adaptive weighting.

**Questions:**

See weakness

---

> ### Author Response · Authors · 2025-11-27
>
> Thank you very much for your thoughtful and encouraging review. We sincerely appreciate your positive feedback on the clarity of our writing and the design of our reward model. We are glad that you found our integration of multiple reward components (correctness, semantics, modality, and format) both coherent and effective, and that our one-stage training framework offered a clear and practical alternative to more complex multi-phase approaches. Your recognition of the motivation behind our LLM-based reward evaluation and overall design choices is deeply appreciated.
>
> ---
>
> ## ***W1. Open-Ended Medical Reinforcement Learning***
>
> Thank you for your comment. While our work builds upon the GRPO RL framework, we found that standard GRPO formulations are not directly applicable to the medical domain, where open-ended, clinically grounded responses are required. Existing RL/GRPO-trained models (e.g., DeepSeek-R1[1], MedVLM-R1[2]) often rely on deterministic rewards such as MCQs or short-form answers. However, practical medical responses rarely fit such rigid formats, as  clinical explanations, symptom descriptions, or diagnostic reasoning can be expressed in diverse yet equally valid ways, even among experts. This makes simple string-matching or single character correctness metrics insufficient and often misleading in evaluating medical responses.
>
> To address this, we extend GRPO to better handle open-ended medical responses under limited training data. We introduce a composite reward system that integrates correctness, semantics, modality grounding, and format compliance. This richer, multi-dimensional reward signal helps reduce reward hacking and stabilizes training issues commonly observed in RL frameworks when the reward structure is overly rigid or sparse. We believe these design and empirical insights constitute meaningful contributions that can generalize to other domains requiring open-ended reasoning.
>
> To further validate the effectiveness of our method, we conducted additional experiments using the default GRPO configuration (i.e., format + accuracy reward with simplified string matching). We found that this default setting, largely optimized for MCQ style or short answer response fails to generalize to open-ended medical responses. Specifically, the constrained reward formulation biases the model toward rigid answer structures, leading to degraded performance on tasks requiring flexible clinical reasoning. While MCQ and short answer accuracies remained consistent or even improved under default GRPO training/reward (showing a 3.22% overall increase over baseline), open ended QA performance deteriorated sharply, with long form response accuracy dropping by 33.86%. This empirical result further highlights the necessity of our proposed reward design for reliably training models that can perform well across both structured and unstructured clinical tasks.
>
> Furthermore, in response to reviewer feedback, we conducted additional experiments using alternative RL frameworks. Our composite reward is effective across multiple RL methods, including DAPO and GSPO. Under the same setting and 1 epoch training, DAPO achieved a score of 60.72%, GRPO attained 59.61%, and GSPO achieved 59.69%, all of them are above the baseline Qwen2.5-VL (57%). These results demonstrate that the proposed composite reward model generalizes well across RL approaches.
>
> Finally, we introduce a unified three-stage evaluation framework combining LLM-based (text-only) and VLM-based (image+text) assessment to more comprehensively benchmark medical multimodal reasoning. This provides a more realistic and holistic evaluation of open-ended clinical performance than existing metrics.
>
> *[1] Guo, Daya, et al. "Deepseek-r1: Incentivizing reasoning capability in llms via reinforcement learning." arXiv preprint arXiv:2501.12948 (2025).*
>
> *[2] Pan, Jiazhen, et al. "Medvlm-r1: Incentivizing medical reasoning capability of vision-language models (vlms) via reinforcement learning." International Conference on Medical Image Computing and Computer-Assisted Intervention. Cham: Springer Nature Switzerland, 2025.*
>
> ---
>
> ## ***W2. On Reproducibility***
>
> All our model weights, training code and curated datasets will be released on open source platforms like Huggingface and Github.
>
> Additionally we have added the python code for our reward functions in the Appendix Sec A.9 .

---

> ### Author Response · Authors · 2025-11-27
>
> ---
>
> ## ***W3. On Reward Weightage***
>
> Thank you for your query. The reward coefficients were selected in the following manner.
>
> All our experiments and ablations were conducted with 10% of the total reward weightage given to the format reward.
>
> Hence, for the ablation experiments shown in Table 4.
>
> - **Embedding only:** *0.1 R_format + 0.9 R_embed*
> - **LLM only:** *0.1 R_format + 0.9 R_llm*
>
> Then, we extended the pipeline by combining both LLM and embedding rewards. For this, we considered the following three variations, all of which maintain the previous 10% reward allocated for the format:
>
> (i) Reward equally split between R_llm and  R_embed
> **v1:** *0.1 R_format + (0.5x0.9)% R_embed + (0.5x0.9)% R_llm*
>
> (ii) More weightage to R_llm
> **v2:** *0.1 R_format + (0.6x0.9)% R_llm + (0.4x0.9)% R_embed*
>
> (iii) More weightage to R_embed
> **v3:** *0.1 R_format + (0.6x0.9)% R_embed + (0.4x0.9)% R_embed*
>
> We observe that the performance does not vary much across the three experiments (v1, v2, v3), achieving average scores of (0.582, 0.589, 0.579), respectively, on our evaluation benchmarks.
>
> For the final version including the modality reward we selected the best version v2 from the previous iteration and allotted 5% weightage to the modality reward, resulting in the final reward weights.
>
> **MediX-R1:** *0.1 R_format + (0.575x0.9)% R_llm + (0.375x0.9)% R_embed + (0.05x0.9) R_modality*
>
> This ensures that the **final reward:** *0.1 + 0.5175 +  0.3375 + 0.045 = 1*
>
> **Please note** that we have NOT performed an exhaustive hyperparameter search for the reward coefficients due to limited computational resources. Instead, the weights were selected manually based on the above ablations and insights from our prior experiments. Nevertheless, this suggests that the excellent results achieved by the proposed method can be further improved by the community through a more comprehensive hyperparameter search, especially as we plan to publicly release our code, dataset, and other resources.

---

### Official Review · Reviewer_9uBU · 2025-11-01

**Soundness:** 2
**Presentation:** 2
**Contribution:** 2
**Rating:** 4
**Confidence:** 3

**Summary:**

This paper introduces MediX-R1, an open-ended reinforcement learning (RL) framework that fine-tunes a medical multimodal model using Group Relative Policy Optimization (GRPO) and a novel composite reward system. This framework employs a multi-signal reward, including an LLM-as-judge accuracy signal and medical embedding-based semantic alignment, along with a unified LLM-based evaluation pipeline, to enable the generation of clinically grounded, free-form answers with interpretable reasoning traces.

**Strengths:**

1. This paper is well-written, logically clear, and easy to follow.
2. The introduced LLM-as-judge evaluation framework effectively addresses, to some extent, the limitations of traditional string-overlap metrics.
3. Extensive experiments demonstrate that the proposed MediX-R1 achieves impressive performance across multiple benchmarks and various tasks.

**Weaknesses:**

1. The source of the $\sim 50$K instruction data used in this paper is missing, which is very important.
2. The composition of the reward weights is very complex, with values including $0.5175$, $0.3375$, and $0.045$. How were these highly precise numerical values derived? Will these weights need to be adjusted as the training data changes?
3. The embedding-based semantic reward uses $0.8$ as the default threshold. Why introduce a threshold to set it as a binary reward instead of directly using the cosine similarity as the reward, which seems to be more precise and can reflect differentiation?
4. Considering that the reasoning process is not supervised during training, how is the reasonableness of the reasoning process inside $\langle\text{think}\rangle\cdots\langle/\text{think}\rangle$ ensured? Is it possible for the chain-of-thought to be flawed or chaotic while the final answer is still correct?
5. It is suggested to mark the best and second-best results for each benchmark in Table 2 (e.g., using bold or underline) to more intuitively reflect the performance comparison.
6. It is recommended to supplement some generated cases from the long-form report generation tasks.

**Questions:**

Please refer to the Weaknesses.

---

> ### Author Response · Authors · 2025-11-27
>
> Thank you very much for your positive and encouraging feedback. We greatly appreciate your recognition of our work’s clarity, the contribution of the LLM-as-judge evaluation framework, and the strong experimental results of MediX-R1. Your thoughtful comments motivate us to continue improving and extending this line of research.
>
> ---
>
> ## ***W1. On Training Data Source***
>
> We apologize for the oversight. The training data was sourced from existing open source medical datasets. The detailed breakdown of our training instructions along with their respective modality is given below. Here, the training samples were prepared by filtering out extremely small images with image height or width less than 28 pixels.
>
> **Training Data:** Total samples: 51,335
>
> **Training Dataset Sources**
>
> | **Dataset**    | **Samples #** |
> |----------------|--------------:|
> | PMC_VQA_SUBSET |         25000 |
> | SLAKE          |          4919 |
> | RAD_VQA        |          1793 |
> | PATH           |         19623 |
> | **Total**      |     **51335** |
>
> **Medical Modality breakdown**
>
> | **Medical Modality** | **Samples #** |
> |:--------------------|--------------:|
> | X_RAY                |         5,964 |
> | MICROSCOPY           |        16,399 |
> | CLINICAL_PHOTOGRAPHY |         8,979 |
> | CT_SCAN              |         7,646 |
> | GRAPHICS             |         2,205 |
> | ANGIOGRAPHY          |           522 |
> | PET_SCAN             |           406 |
> | ULTRASOUND           |         1,227 |
> | MRI_SCAN             |         6,224 |
> | FUNDUS_PHOTOGRAPHY   |           314 |
> | OCT_SCAN             |           236 |
> | ENDOSCOPY            |           611 |
> | MAMMOGRAPHY          |           106 |
> | FLUOROSCOPY          |           321 |
> | OTHER                |            64 |
> | SPECT                |           111 |
> | **Total**            |    **51,335** |
>
> Our training dataset has been curated from the corresponding train split of the aforementioned  source datasets.
>
> We have **added these tables to the Appendix Sec A.7** of the revised paper. Additionally, the **training configuration is added to Appendix Sec A.8.**
>
> ---
>
> ## ***W2. On Reward Weightage***
>
> Thank you for your query. The reward signal was designed in the following manner.
>
> All our experiments and ablations were conducted with 10% of the total reward weightage given to the format reward.
>
> Hence, for the ablation experiments shown in Table 4.
>
> - **Embedding only:** *0.1 R_format + 0.9 R_embed*
> - **LLM only:** *0.1 R_format + 0.9 R_llm*
>
> Then, we extended the pipeline by combining both LLM and embedding rewards. For this, we considered the following three variations, all of which maintain the previous 10% reward allocated for the format:
>
> (i) Reward equally split between R_llm and  R_embed
> **v1:** *0.1 R_format + (0.5x0.9)% R_embed + (0.5x0.9)% R_llm*
>
> (ii) More weightage to R_llm
> **v2:** *0.1 R_format + (0.6x0.9)% R_llm + (0.4x0.9)% R_embed*
>
> (iii) More weightage to R_embed
> **v3:** *0.1 R_format + (0.6x0.9)% R_embed + (0.4x0.9)% R_embed*
>
> We observe that the performance does not vary much across the three experiments (v1, v2, v3), achieving average scores of (0.582, 0.589, 0.579), respectively, on our evaluation benchmarks.
>
> For the final version including the modality reward we selected the best version v2 from the previous iteration and allotted 5% weightage to the modality reward, resulting in the final reward weights of
>
> **MediX-R1:** *0.1 R_format + (0.575x0.9)% R_llm + (0.375x0.9)% R_embed + (0.05x0.9) R_modality*
>
> This ensures that the final reward: 0.1 + 0.5175 +  0.3375 + 0.045 = 1
>
> Please note that we have NOT performed an exhaustive hyperparameter search for the reward coefficients due to limited computational resources. Instead, the weights were selected manually based on the above ablations and insights from our prior experiments. Comparable performance of v1, v2, and v3 indicates the robustness of the loss weights, which validates that it is not required to change the loss weights when the training data changes.  Nevertheless, we hope that the results can be further enhanced  by the community through a more comprehensive hyperparameter search, especially as we plan to publicly release our code, dataset, and other resources.

---

> ### Author Response · Authors · 2025-11-27
>
> ---
> ## ***W3. On Embedding Reward***
>
> Thank you for your suggestion. We appreciate the reviewer’s insight regarding the use of a threshold based binary reward versus a continuous cosine similarity signal. As described in the DeepSeek-R1[1] paper, the GRPO framework relies primarily on rule based accuracy rewards that evaluate correctness in a binary manner, particularly for deterministic tasks such as math and programming problems. This design choice helps mitigate reward hacking, which DeepSeek authors identified as a key risk when employing neural based reward models that assign partial or continuous scores. Moreover, defining reliable fine grained intermediate rewards in reasoning tasks remains challenging and often destabilizes training.
>
> However following your suggestion we explored a non-binary variant, where the reward equaled the cosine similarity if cosine ≥ 0.8, and 0 otherwise. In comparison with the binary approach, both variants achieved comparable performance (0.610-binary vs. 0.602-continuous). However, the binary reward yielded slightly more stable learning dynamics and marginally better convergence. Consistent with findings from the DeepSeek-R1 framework, we attribute this to reduced reward noise and a lower risk of reward hacking.
>
> *[1] Guo, Daya, et al. "Deepseek-r1: Incentivizing reasoning capability in llms via reinforcement learning." arXiv preprint arXiv:2501.12948 (2025).*
>
> ---
>
> ## ***W4. On Reasoning Evaluation***
> Thank you for this insightful question. We acknowledge that, similar to the findings reported in the DeepSeek-R1[1] paper, models trained purely through reinforcement learning (e.g., DeepSeek-R1-Zero) can indeed produce correct final answers despite exhibiting chaotic or flawed reasoning traces. The DeepSeek-R1 authors explicitly note that R1-Zero achieved strong reasoning benchmark performance but suffered from poor readability and language mixing, indicating imperfect intermediate reasoning despite accurate outputs. In our case, MediX-R1 employs a composite reward system designed to better accommodate open-ended medical question answering, where correctness cannot be verified by a single deterministic or verifiable token. Our assumption is that generating a clinically coherent and contextually complete answer requires the model to attend effectively to its preceding reasoning traces. Because the answers in our setup are typically long and open-ended, the reasoning process must remain consistent and well grounded throughout many tokens, which naturally reduces the likelihood of flawed or chaotic reasoning. In contrast, tasks with short, single token verifiable rewards provide limited opportunity for such internal consistency.
>
> To further validate this, we have extended our human expert study to evaluate the reasoning quality of our MediX-R1 model against MedGemma with the help of 5 medical doctors. Experts assessed outputs for clinical accuracy, reasoning soundness, and practical usefulness in a medical setting. MediX-R1’s reasoning was preferred in 74.2% of cases over MedGemma, indicating stronger clinical coherence.  Furthermore, the study shows that in 92.4% of the cases, the model’s reasoning steps were rated as acceptable and often comparable to a medical doctor’s thought process, while only 7.6% of the cases were rated as having poor reasoning quality. Moreover, in fewer than 5% of the cases, the model produced flawed reasoning despite generating the correct final answer, indicating that such inconsistencies are rare and that MediX-R1 generally maintains a robust and coherent reasoning process. These details are updated in Sec A.6 of the revised paper.
>
> *[1] Guo, Daya, et al. "Deepseek-r1: Incentivizing reasoning capability in llms via reinforcement learning." arXiv preprint arXiv:2501.12948 (2025).*
>
> ---
>
> ## ***W5. Table 2 - Bold and Underline results***
>
> Thank you for your suggestion. We have updated Table 2 in the revised paper to account for this.
>
> ---
>
> ## ***W6. Report Generation Samples***
>
> Thank you for your suggestion. We have added sample cases for long form report generation in the Appendix Sec A.5 of the revised paper

---

### Author Response · Authors · 2025-12-04
**Summary of Revisions and Clarifications**

*Dear Reviewers,*

Thank you for your thoughtful and constructive feedback. We are encouraged by the consistently positive assessments across reviewers, who highlighted the paper as *“well-written,”* *“clearly structured,”* and *“easy to follow”* (9uBU, Z9Lo) and praised our contribution to the timely area of RL-based medical reasoning (jMAA). Reviewers recognized the proposed composite reward as *“thoughtful,”* and *“well-motivated,”* effectively combining correctness, semantics, modality, and format signals in a way that stabilizes open-ended medical RL (Z9Lo, MGH6, jMAA). The unified LLM-as-judge framework was noted as a meaningful improvement over traditional string-overlap metrics and a consistent, semantically aware evaluator across modalities (9uBU, MGH6). Reviewers also commended the model’s broad modality coverage and strong empirical performance across diverse benchmarks (9uBU, MGH6), as well as the simplicity and effectiveness of our one-stage RL design compared to more complex multi-phase pipelines (Z9Lo).

**Open-Ended Medical RL & Composite Reward**

The key contribution of our method lies in enabling RL to work for open-ended clinical reasoning where prior RL systems generally fail. With this objective,  we introduce a simple, yet effective  composite reward (LLM correctness, semantic similarity, modality grounding, and format) plus a unified evaluator pipeline. Unlike default GRPO, which boosts MCQs but hurts long-form QA, our reward improves various task types and remains effective across GRPO, DAPO, and GSPO. Furthermore, our approach is generalizable across multiple backbones, and we achieve consistent performance gains on Qwen2.5-VL, Qwen3-VL, and SmolVLM2 backbones.

**LLM-as-Judge Reliability & Human Expert Validation**

We strengthened evaluation reliability by adding deterministic multi-run checks with ±0.002 variance, cross-model agreement across Qwen3-14B, GPT-5.1, GPT-5 mini judges, and explaining why binary judgments outperform string metrics (see Reviewer jMAA W2). We also detailed a human expert study with 5 certified clinicians with 63% agreement between them; MediX-R1’s reasoning was preferred in 74.2% of cases, with 92.4% judged clinically acceptable. This directly addresses concerns about evaluator robustness and clinical validity.

**Generalization to New Backbones & Real-World Clinical Data**

To address generalizability, we demonstrate that applying our composite rewards to the stronger Qwen3-VL backbone yields an additional \~2% gain. In addition, we experimented with our composite rewards on SmolVLM2 backbone, and observed similar gains. These consistent gains across diverse model families demonstrate that MediX-R1 enhances reasoning ability beyond model-specific reward tuning.

**RL Algorithms, Compute Efficiency, and Simplicity**

We expanded comparisons across RL methods: with the same reward framework, DAPO (60.72%), GRPO (59.61%), and GSPO (59.69%) all exceed the baseline (57%). MediX-R1 trains in a single stage on 8×A100s for \~25 hours roughly half the time of multi-stage baselines like BiMediX2 (\~52 hours). This supports reviewer Z9Lo’s observation that our framework maintains strong performance with a simpler and more efficient design.

**Data Transparency, Training Details, and Reproducibility**

We now fully describe the 51,335 sample training corpus (modalities, sources, filtering), configuration details (Sec. A.7-A.8), and provide composite reward function code (Sec. A.9). We also improved presentation (highlighted tables, qualitative outputs) and commit to releasing all model weights, datasets, and full training/evaluation code after the review period addressing reproducibility concerns from Z9Lo, 9uBU, and jMAA.

We have also uploaded a revised draft addressing almost all concerns raised by the reviewers. We thank the reviewers for their constructive suggestions, which have helped us further strengthen the clarity and impact of our work. Please let us know if any additional clarification would be helpful.

*Yours Sincerely,*

*Authors of MediX-R1*

---

> ### Author Response · Authors · 2025-12-04
> **List of Revisions**
>
> ## **List of Revisions**
>
> **Reviewer 9uBU**
>
> **Strengths:**
>
> - The paper is well-written, clear, and easy to follow.
> - The LLM-as-judge evaluation framework mitigates limitations of traditional string-overlap metrics.
> - Extensive experiments show that MediX-R1 performs strongly across benchmarks and tasks.
>
> **Weaknesses:**
>
> **W1. On Training Data Source:** Training Data sources and modality breakdown added. Data sources and training configuration updated in Appendix Sec A.7 and A.8
>
> **W2. On Reward Weightage:** The composition of reward co-efficients and intermediate experiments are detailed in the response.
>
> **W3. On Embedding Reward:** Additional experiment with non-binary embedding reward is added in the response.
>
> **W4. On Reasoning Evaluation:** Extended our human expert study to evaluate model reasoning quality. Results are added in Appendix Sec A.6.
>
> **W5. Table 2 - Bold and Underline result:** Table 2 is updated to account for the changes.
>
> **W6. Report Generation Samples:** Report generation samples are added in Appendix Sec A.5
>
> ---
>
> **Reviewer Z9Lo**
>
> **Strengths:**
> - The paper is well-written and clearly structured, making complex RL ideas easy to follow.
> - It introduces a well-designed reward model that integrates complementary signals (correctness, semantics, modality, format).
> - The one-stage training setup keeps the framework simple and effective, avoiding unnecessary multi-phase and agentic complexity.
>
> **Weaknesses:**
>
> **W1. Open-Ended Medical Reinforcement Learning:** Addressed the need for medically aligned reward framework for GRPO; Additional experiments on Default GRPO configuration and RL methods - DAPO, GRPO, GSPO - shows consistent gains.
>
> **W2. On Reproducibility:** Reward function code added to Appendix Sec A.9. Full training code and model weights will be uploaded to Github and HuggingFace.
>
> **W3. On Reward Weightage:** The composition of reward co-efficients and intermediate experiments are detailed in the response.
>
> ---
>
> **Reviewer MGH6**
>
> **Strengths:**
> - Well-motivated multi-signal reward design (LLM, embeddings, modality, format) that stabilizes open-ended RL.
> - Unified LLM-as-judge evaluation offering consistent, semantically aware assessment across text and image tasks.
> - Broad modality support (CT, X-ray, MRI, microscopy, ultrasound, etc.) that enhances clinical versatility.
>
> **Weaknesses:**
>
> **W1. Results on alternative backbones:** Additional results on backbones like Qwen3-VL and SmolVLM2, shows consistent gains with our composite rewards.
>
> **W2. On Human Expert Evaluation:** Extended our human expert study to evaluate model reasoning quality, added agreement and doctor qualifications.
>
> **W3. Open-Ended Medical Reinforcement Learning Contributions:** Addressed the need for medically aligned reward framework for GRPO; Additional experiments on Default GRPO configuration and RL methods - DAPO, GRPO, GSPO - shows consistent gains.
>
> **Q1. On Training Compute:** Compute Budget added and compared with BiMediX2. Training configuration added to Appendix Sec. A.8
>
> **Q2. Evaluation on real-world Clinical data:** Additional evaluation of real world hospital data from MedPiX2.0. MediX-R1 outperforms other medical VLMs in this evaluation. Results added to Appendix Sec A.10.
>
> **Q3. Results on alternative backbones:** Additional results on backbones like Qwen3-VL and SmolVLM2, shows consistent gains with our composite rewards.
>
> ---
>
> **Reviewer jMAA**
>
> **Strengths:**
> - Addresses a timely topic on RLVR-based medical reasoning amid the growing interest in RLVR for text-only tasks.
> - Proposes a structured four complimentary component reward design useful for future medical RLVR and multimodal RL research.
>
> **Weaknesses:**
>
> **W1. Open sourcing code and models & Open-Ended Medical Reinforcement Learning Contributions:** Addressed the need for medically aligned reward framework for GRPO; Additional experiments on Default GRPO configuration and RL methods - DAPO, GRPO, GSPO - shows consistent gains. Reward function code added to Appendix Sec A.9. Full training code and model weights will be uploaded to Github and HuggingFace.
>
> **W2. On LLM-as-a-Judge:** Additional evaluation results using GPT-5.1 and GPT-5 mini as evaluators are performed. Extended our human expert study to evaluate model reasoning quality, added agreement and doctor qualifications.
>
> **W3. On different RL Algorithms and backbones:** Additional experiments on RL methods beyond GRPO, shows consistent gains across DAPO, GRPO, GSPO. Additional results on backbones like Qwen3-VL and SmolVLM2, shows consistent gains with our composite rewards.
>
> **W4. On Training Data Source:** Training Data sources and modality breakdown added. Data sources and training configuration updated in Appendix Sec A.7 and A.8
>
> **Q1. On Human Expert Evaluation:** Extended our human expert study to evaluate model reasoning quality, added agreement and doctor qualifications.
>
> **Q2. On training dataset details:** Addressed in W4.
>
> ---

---

### Meta-Review · Area_Chair_qmc1 · 2026-01-04

**Summary:**

Reviewers biggest concerns center on a few key points: (1) Novelty: The reviewers thought the method resembles many others and they found it largely incremental. While the authors list some technical differences from alternatives, the reviewers didn't find the contribution to be clear and large-enough. (2): Experimental Rigor: The reviewers found the experiments too reliant on LLM-as-a-judge, and found the experiments didn't fully isolate the source of performance via hyperparameter studies/ablations on the proposed methods. These issues are partially resolved through human validations and more descriptions of the hyperparameters, but deserve further details and experiments in the paper to help future readers understand when and why the proposed method is expected to work. (3) Missing Experimental Details: The reviewers had confusion around specifics of some of the experiments. The authors provide these details and the paper can be made clearer accordingly, so this is not a major concern.

**Reviewer Concerns:**

**Addressed Concerns**:
* Many of the clarity concerns can be addressed in the writing, as demonstrated by the responses
* Some of the experimental sufficiency issues have also already been addressed and should be expanded where possible and included in the updated paper.
* Open-sourcing the code.
* Added more backbones

**Unaddressed Concerns**:
* The proposed method isn't a clear, conceptual improvement beyond recent alternatives. This is a hazard of working in popular topics, so it's hard to address. But expanding the exploration of the solution space via a hyperparameter study alongside further ablations will help.

**Reviewer Scores:**

R1: Possible increase due to increased clarity

R2, R3, R4: Likely unchanged due to novelty concerns.

---

### Decision · Program_Chairs · 2026-01-26

Reject